# An IL-17-EGFR-TRAF4 axis contributes to the alleviation of lung inflammation in severe influenza

Avijit Dutta [1,2], Chen-Yiu Hung[3], Tse-Ching Chen [4,5], Sung-Han Hsiao[2], Chia-Shiang Chang[2], Yung-Chang Lin[6,7], Chun-Yen Lin [8,9] & Ching-Tai Huang [2,10 ✉]

Excessive inflammation is a postulated cause of severe disease and death in respiratory virus infections. In response to severe influenza virus infection, adoptively transferred naïve hemagglutinin-specific $CD4^+$ T cells from $CD4^+$ TCR-transgenic 6.5 mice drive an IFN-γ-producing Th1 response in wild-type mice. It helps in virus clearance but also causes collateral damage and disease aggravation. The donor 6.5 mice have all the $CD4^+$ T cells with TCR specificity toward influenza hemagglutinin. Still, the infected 6.5 mice do not suffer from robust inflammation and grave outcome. The initial Th1 response wanes with time, and a prominent Th17 response of recent thymic emigrants alleviates inflammation and bestows protection in 6.5 mice. Our results suggest that viral neuraminidase-activated TGF-β of the Th1 cells guides the Th17 evolution, and IL-17 signaling through the non-canonical IL-17 receptor EGFR activates the scaffold protein TRAF4 more than TRAF6 during alleviation of lung inflammation in severe influenza.

[1] Research Center for Emerging Viral Infections, College of Medicine, Chang Gung University, Guishan-33302, Taoyuan City, Taiwan. [2] Division of Infectious Diseases, Department of Medicine, Chang Gung Memorial Hospital, Guishan-33333, Taoyuan City, Taiwan. [3] Division of Thoracic Medicine, Department of Medicine, Chang Gung Memorial Hospital, Guishan-33333, Taoyuan City, Taiwan. [4] Department of Pathology, Chang Gung Memorial Hospital, Guishan-33333, Taoyuan City, Taiwan. [5] Department of Pathology, College of Medicine, Chang Gung University, Guishan-33302, Taoyuan City, Taiwan. [6] Division of Hematology and Oncology, Department of Medicine, Chang Gung Memorial Hospital, Guishan-33333, Taoyuan City, Taiwan. [7] Division of Hematology and Oncology, College of Medicine, Chang Gung University, Guishan-33302, Taoyuan City, Taiwan. [8] Division of Hepatogastroenterology, Department of Medicine, Chang Gung Memorial Hospital, Guishan-33333, Taoyuan City, Taiwan. [9] Division of Hepatogastroenterology, College of Medicine, Chang Gung University, Guishan-33302, Taoyuan City, Taiwan. [10] Division of Infectious Diseases, College of Medicine, Chang Gung University, Guishan-33302, Taoyuan City, Taiwan. ✉email: chingtaihuang@gmail.com

Respiratory viruses cause serious emerging infectious diseases. The clinical outcome depends on the interplay between the invading virus and the host immune response. Viral invasion causes a cytopathic effect that leads to tissue injury in the respiratory tract. The host immune system reacts to clear the virus, but the response also contributes to inflammation. If the immune activation and inflammation abate with the virus clearance, the infected individual recovers efficiently from an infection. However, immune activation and inflammation may persist beyond the virus clearance. Prolonged and exaggerated immune activation causes aggravation of the disease. The undesirable outcome of seasonal or pandemic influenza virus infections, SARS, MERS, and the ongoing COVID-19 pandemics are examples[1]. Modulation of the inflammation with overwhelming immune reactions is always a goal in disease management. Corticosteroid as a commonly implemented agent to quell lung inflammation also impairs pathogen clearance with suppressed inflammation and may aggravate the disease, as seen in severe influenza[2–5], SARS[6], MERS[7], and COVID-19 patients[8].

Targeting cytokines is another option for suppressing inflammation. Tocilizumab and sarilumab target interleukin (IL)−6, and may help COVID-19 patients[9]. IL-17 is also a well-known pro-inflammatory cytokine[10]. The high IL-17 levels in patients with severe diseases of COVID-19[1,11,12], SARS[13], MERS[14], and influenza[15], make IL-17 a possible target of manipulation[16]. We detected IL-17 in the sera of our patients with severe COVID-19[11], and also in lung-infiltrating influenza antigen-specific CD4+ T cells in our experimental mouse model of severe influenza[17,18].

In our influenza hemagglutinin (HA) antigen-specific mouse model, the influenza virus drives a Th1 response of adoptively transferred naïve HA-specific CD4+ T cells in the lungs[17–22]. The $2.5 \times 10^6$ adoptively transferred donor cells help in virus clearance but boost inflammation in the lungs and enhance mortality. Influenza virus infection of the donor mice, the HA-specific CD4+ T cell receptor (TCR)-transgenic 6.5 mice[23], drives activation of all CD4+ T cells in these mice, as all their CD4+ T cells have the TCRs specific to the HA of the influenza virus. We expected a grave outcome with such a massive Th1 response of all CD4+ T cells in the infected 6.5 mice. Although they lost more body weight at first, they suffered from mild lung inflammation and recovered well. Their recovery was associated with a prominent Th17 response on days 6 and 9. Here we studied the role of Th17 response in protection of mice against severe influenza. We demonstrated a Th1 guided evolution of Th17 response. Our results suggest that viral neuraminidase-activated TGF-β of the Th1 cells guides the Th17 evolution. IL-17 signaling through the non-canonical IL-17 receptor EGFR activates the scaffold protein TRAF4 more than TRAF6 during alleviation of lung inflammation in severe influenza.

## Results

### Th17 response is associated with the protection against severe influenza

In our influenza hemagglutinin (HA) antigen-specific mouse model[17–22], wild-type mice received adoptive transfer of $2.5 \times 10^6$ naïve HA-specific 6.5 CD4+ T cells from 6.5 donor mice (WT mice + 6.5 cells) and got $2.5 \times 10^3$ plaque-forming units (p.f.u.) PR8 strain of H1N1 influenza virus infection. Lung-infiltrating HA-specific 6.5 CD4+ T cells (donor 6.5 cells) produced IFN-γ with the induction of Th1 transcription factor T-bet (Fig. 1a). There was base level IL-17 production with negligible Th17 transcription factor ROR-γt (Fig. 1a, Supplementary Note 1, Supplementary Fig. 1). Adoptive transfer aggravated the disease of wild-type mice with influenza virus infection. Lungs were inflamed (Fig. 1b), and lung-infiltrating donor 6.5 cells acquired effector function on day 6. These cells augmented the HA peptide-stimulated proliferation of cognate naïve CD4+ T cells

in vitro (Fig. 1b). Virus burden also decreased in the lung from day 3 to day 9, as revealed in the plaque assay (Fig. 1c). The mice continued to lose body weight up to day 8, when some of them started to die (Fig. 1c).

The donor mice of the naïve HA-specific 6.5 CD4+ T cells, the 6.5 TCR transgenic mice, have monotonous TCR on their CD4+ T cells, with the same α/β TCR responding to an MHC Class II I-E$^d$ restricted epitope of the HA from A/PR/8/34 (PR8) strain H1N1 influenza virus[23]. We expected a grave outcome of influenza virus infection in the donor 6.5 mice, with a massive Th1 response of all CD4+ T cells in the infected 6.5 mice. In contrast to our anticipation, the prominent Th1 response with IFN-γ production was noted on day 3 only, but not thereafter on days 6 and 9 after $2.5 \times 10^3$ p.f.u. PR8 virus infection (Fig. 1a). On days 6 and 9, with activation of Th17 transcription factor ROR-γt in 30 ~ 50% of the lung-infiltrating CD4+ T cells, 25 ~ 40% of them produced IL-17 instead (Fig. 1a, Supplementary Note 1, Supplementary Fig. 2). Infected 6.5 mice displayed alleviated lung inflammation (Fig. 1b), and their lung-infiltrating CD4+ T cells acquired regulatory function on day 6. These cells suppressed the HA peptide-stimulated proliferation of naïve cognate CD4+ T cells in vitro (Fig. 1b). Infected 6.5 mice lost significant body weight on day 3. They recovered from weight loss and all mice survived the infection (Fig. 1c). There was no live virus in the lungs on day 9 (Fig. 1c).

IL-17 (interleukin-17A) is a well-claimed pro-inflammatory cytokine which contributes to the pathogenesis of many inflammatory diseases[11–16]. But, a prominent IL-17 response of lung-infiltrating CD4+ T cells is associated with alleviated lung inflammation here in the infected 6.5 mice. The alleviation of lung inflammation and protection from influenza virus infection was lost in IL-17 knockout (KO) 6.5 mice. With $2.5 \times 10^3$ p.f.u. PR8 influenza virus, IL-17KO 6.5 mice incurred severe disease. The lung-infiltrating IL-17KO CD4+ and CD8+ T cells produced more inflammatory cytokines IFN-γ and TNF-α on day 6 (Fig. 1d).

### Th17 is the response of recent thymic emigrants

Activated CD4+ T cells expand at first, followed by contraction of the pool of cells. Influenza virus-driven activation of the adoptively transferred naïve HA-specific 6.5 CD4+ T cells also followed a similar trend in infected wild-type mice (WT mice + 6.5 cells) during our experiments. The donor 6.5 cell pool expanded initially and then contracted in the lungs of wild-type mice (Fig. 2a). In contrast, the lung-infiltrating CD4+ T cell population decreased in the infected 6.5 mice in the first 4 days. It then started to increase and return to the level prior to infection by day 11 post-infection (Fig. 2a). Cells from the thymus of 6.5 mice, the new thymic emigrants, may replenish the population of lung-infiltrating CD4+ T cells. We thus tested CD24 and Qa2 expression profile of the lung-infiltrating CD4+ T cells in infected 6.5 mice. The matured thymocytes are known to express high-level CD24 and minimal Qa2. The CD4+ or CD8+ T cells lose CD24 and gain Qa2 expressions with time after being released to the periphery[24]. Among the lung-infiltrating CD4+ T cells of the infected 6.5 mice, we found a higher CD24 and lower Qa2 expressions on the surface of IL-17-producing than the IFN-γ-producing cells on day 6 (Fig. 2a).

To validate the above finding that suggests the recent thymic emigrants as the source of IL-17 in the lungs of infected 6.5 mice, we did thymectomy in 2-week-old 6.5 mice and infected the thymectomized 6.5 mice after 2 weeks of recovery from the surgery. Thymectomy is the surgical removal of the thymus gland. It thus abolishes the supply of thymic emigrants. After $2.5 \times 10^3$ p.f.u. PR8 influenza virus infection, thymectomized 6.5 mice had less ROR-γt induction and IL-17 production in the lung-infiltrating CD4+ T cells on day 7 (Fig. 2b). The proportions of IFN-γ-producing cells

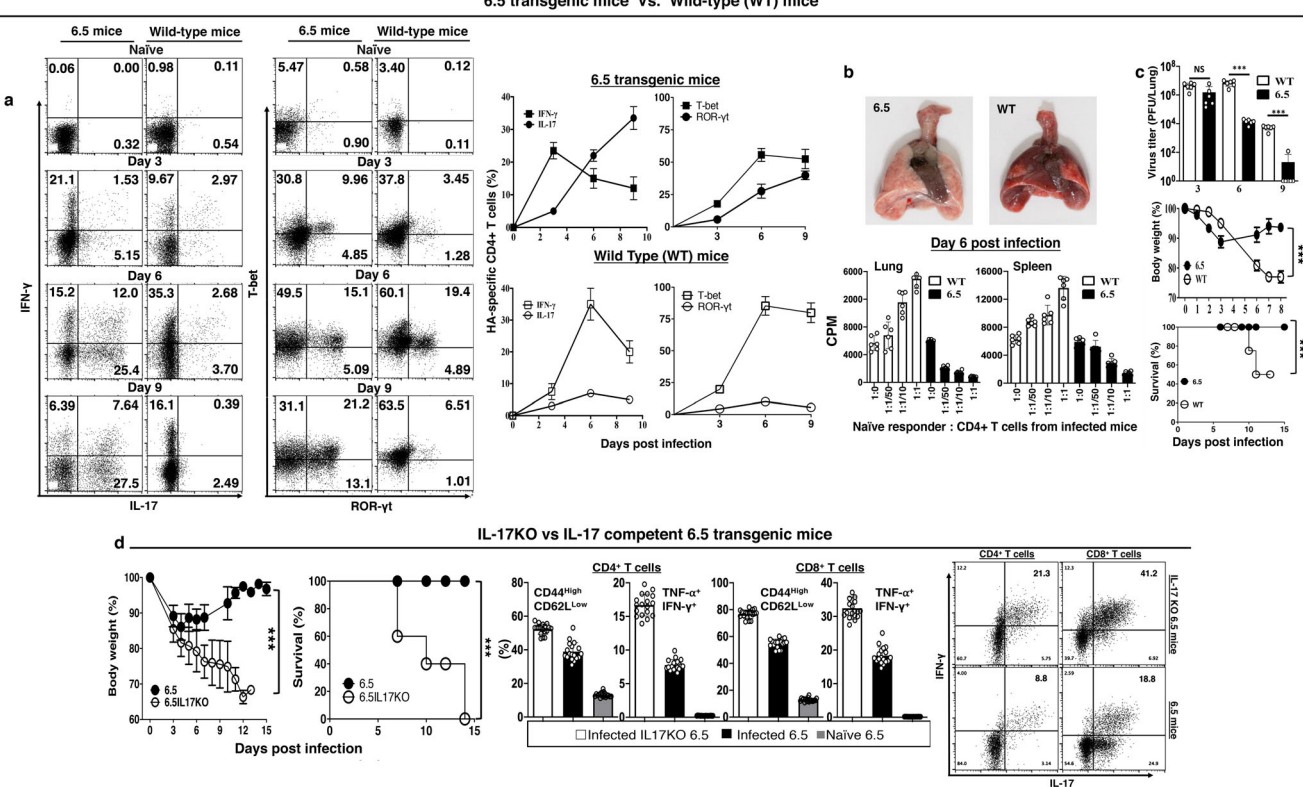

**Fig. 1 IL-17 response of lung-infiltrating CD4+ T cells in TCR-transgenic 6.5 mice is associated with improved recovery and survival benefit against severe influenza. a–c** *Th17 response and alleviated lung inflammation in HA-specific CD4+ TCR-transgenic 6.5 mice.* **a** Th1 and Th17 responses of lung-infiltrating HA-specific CD4+ T cells, **b** Upper panels: gross lung pathology on day 6; Lower panels: naïve HA-specific CD4+ T cell proliferation in vitro with infection-responding cells from lungs or spleens (**c**) virus burden in the lungs, body weight loss, and survival. Wild type (WT) non-transgenic mice received adoptive transfer of 2.5 × 10⁶ naïve HA-specific CD4+ T cells with infection, as described in text, and served as controls. **d** *Lost survival benefit with IL-17 deficiency in TCR-transgenic 6.5 mice.* Severe bodyweight loss and mortality with more inflammatory cytokine secretion by the lung-infiltrating CD4+ and CD8+ T cells in infected IL-17KO than IL-17-competent TCR-transgenic 6.5 mice. Photographs are representatives, and other values are means ± s.d. of at least 3 experiments (n = 6/group; ***p < 0.0001; NS non-significant, p > 0.05; two-tailed P values for unpaired t-test to compare two groups at a specific time point, Two-way ANOVA for body weight curves, and Gehan–Breslow–Wilcoxon test for survival curves).

were not changed by thymectomy, and ~15% of the lung-infiltrating CD4+ T cells produced IFN-γ in the thymectomized and non-thymectomized 6.5 mice by this time (Fig. 2b). As a result, thymectomy obliterated the IL-17 dominance in infected 6.5 mice and reverted the distribution between IL-17 and IFN-γ back to the pattern similar to that of donor 6.5 cells in infected wild-type mice (Fig. 2b). With this altered Th response, thymectomy caused higher virus burden in the lungs (Fig. 2c), and aggravated disease with more bodyweight loss (Fig. 2d) in the infected 6.5 mice. These results corroborate our findings that suggest Th17 is a protective response of recent thymic emigrants in infected 6.5 mice.

**Waning Th17 with aging-related thymic involution and severe disease of influenza virus infection in older than younger adult mice.** Thymic involution causes decreased supply of thymic emigrants with age. Similar to wild-type mice, there were about 50 million thymocytes in 4-week-old 6.5 mice. The number decreased to 40 million in 8-week-old and 20 million in 12-week-old 6.5 mice. There were only 10 million thymocytes in 40-week-old 6.5 mice (Fig. 3a). In addition, there were less CD4+CD8+ double-positive cells in CD3+ thymocytes with age (Fig. 3b). Influenza virus infection induced less Th17 response in aged 6.5 mice compared to young 6.5 mice. About 30% of lung-infiltrating CD4+ T cells produced IL-17 in 4 or 8 week-old 6.5 mice on day 7 after the infection with 2.5 × 10³ p.f.u. of PR8 influenza virus. The number decreased to less than 20% in 12-week-old and to

around 10% in 40-week-old 6.5 mice (Fig. 3c, d). However, the percentage of IFN-γ producing cells did not change with age, and about 10–12% lung-infiltrating CD4+ T cells produced IFN-γ in the 4–40-week-old 6.5 mice (Fig. 3c, d). As the percentage of IFN-γ producing cells did not change with age, the IL-17 dominance over IFN-γ production also decreased with age (Fig. 3e). Virus burden in the lungs was higher in old than young 6.5 mice on day 7 after infection (Fig. 3f). Old 6.5 mice suffered from disease of more severity than young 6.5 mice, with delayed recovery of the weight loss (Fig. 3g).

**The evolution of Th1-guided Th17 response.** We devised an experiment with two adoptive transfers of naïve HA-specific 6.5 CD4+ T cells into infected wild-type mice (WT mice + 2-batches of 6.5 cells) to mimic sequential thymic emigrants in 6.5 mice (Fig. 4a). The donor 6.5 cells transferred with infection differentiated into IFN-γ-producing Th1 cells, and the donor 6.5 cells transferred 4 days later differentiated into IL-17-producing Th17 cells on day 8 after infection in the lungs of wild-type mice (Fig. 4a–c, Supplementary Note 2, Supplementary Fig. 3). In the presence of Th1 cells derived from first transfer, 30% of the second transfer cells produced IL-17 and only less than 2% produced IFN-γ. About 5% of IL-17-producing cells produced IFN-γ as well (Fig. 4d). The second transfer cells also modified the phenotype of first transfer cells. There was a 40% reduction of IFN-γ production on day 8 after infection (Fig. 4d). In single

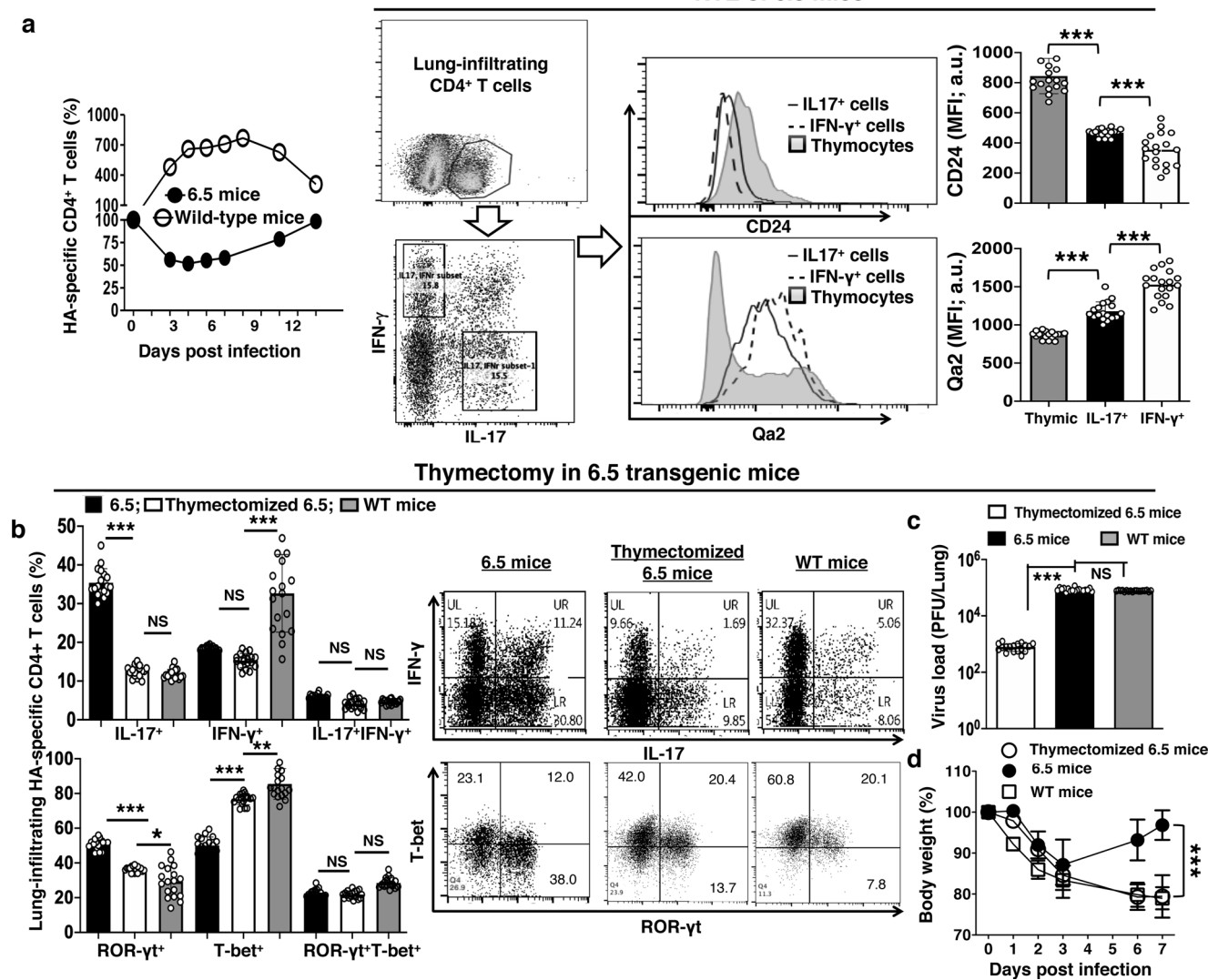

**Fig. 2 Th17 is the response of recent thymic emigrants in TCR-transgenic 6.5 mice. a** Recent thymic emigrants secrete more IL-17 and less IFN-γ in the infected lungs of 6.5 mice: Left panel: lung-infiltrating HA-specific CD4+ T cell population kinetics in infected 6.5 mice. WT mice with adoptively transferred cells served as controls, as described in text. Middle and right panels: CD24 and Qa2 expression profiles of IL-17 or IFN-γ producing lung-infiltrating CD4+ T cells in infected 6.5 mice. **b–d** Thymectomy abolishes the supply of recent thymic emigrants and thymectomized 6.5 mice suffer from severe disease with less IL-17 production: **b** IL-17 and IFN-γ production with ROR-γt and T-bet induction of lung-infiltrating HA-specific CD4+ T cells and **c** Virus burden in the lungs on day 7. **d** Body weight loss after infection. Non-thymectomized 6.5 mice served as controls. WT mice with adoptively transferred cells served as another group of control mice, as described in text. Histograms and dot-plots are representatives, and other values are means ± s.d. of at least 3 experiments (n = 6/group; ***p < 0.0001; two-tailed P values for unpaired t-test to compare two groups at a specific time point, Two-way ANOVA for body weight curves).

transfer controls (WT mice + single-batch 6.5 cells), the donor 6.5 cells transferred with infection or on days 4 post infection were prominent IFN-γ-producing Th1 cells on day 8 post infection in the lungs of wild-type mice. There was only a marginal decrease of IFN-γ production and a marginal increase of IL-17 production of the day 4 transferred cells compared to the cells transferred with infection. (Fig. 4).

In the experiment of wild-type mice with two adoptive transfers of naïve HA-specific 6.5 CD4+ T cells (WT mice + 2-batches of 6.5 cells), the IL-17-producing Th17 response of the second transfer donor 6.5 cells was associated with a decreased dominance of T-bet over ROR-γt (Fig. 5a, b). There were comparable TCR signal transduction from ZAP-70 phosphorylation to the induction of hallmark Th1 transcription factor T-bet in donor 6.5 cells of both transfers (Fig. 5c). With the substantial T-bet induction, there was a strong activation of Th17-associated transcription factor ROR-γt in

the donor 6.5 cells of the second transfer (Fig. 5d). The donor 6.5 cells of the second transfer produced IL-17 instead of IFN-γ with the reduced T-bet dominance over ROR-γt (Fig. 5b–d). In addition, there was higher expression of the induced regulatory T (iTreg) cell marker LAG-3 in the donor 6.5 cells of the second transfer (Fig. 5e).

The two adoptive transfers of donor 6.5 cells helped the recipient wild-type mice to recover from the infection more efficiently with better survival, compared to the mice with one transfer of donor 6.5 cells only (Fig. 5f). Improved protection was there with less pathology (Fig. 5g), and lower viral load in the lungs on day 8 after infection (Fig. 5h).

We further defined the protective role of IL-17 from the second batch cells in the experiment with two adoptive transfers, by using donor 6.5 cells from naïve IL-17KO 6.5 mice in the second adoptive transfer ((WT mice + 1st batches 6.5 cells + 2nd batch IL-17KO 6.5 cells; Fig. 6a). Replacement of the second transfer

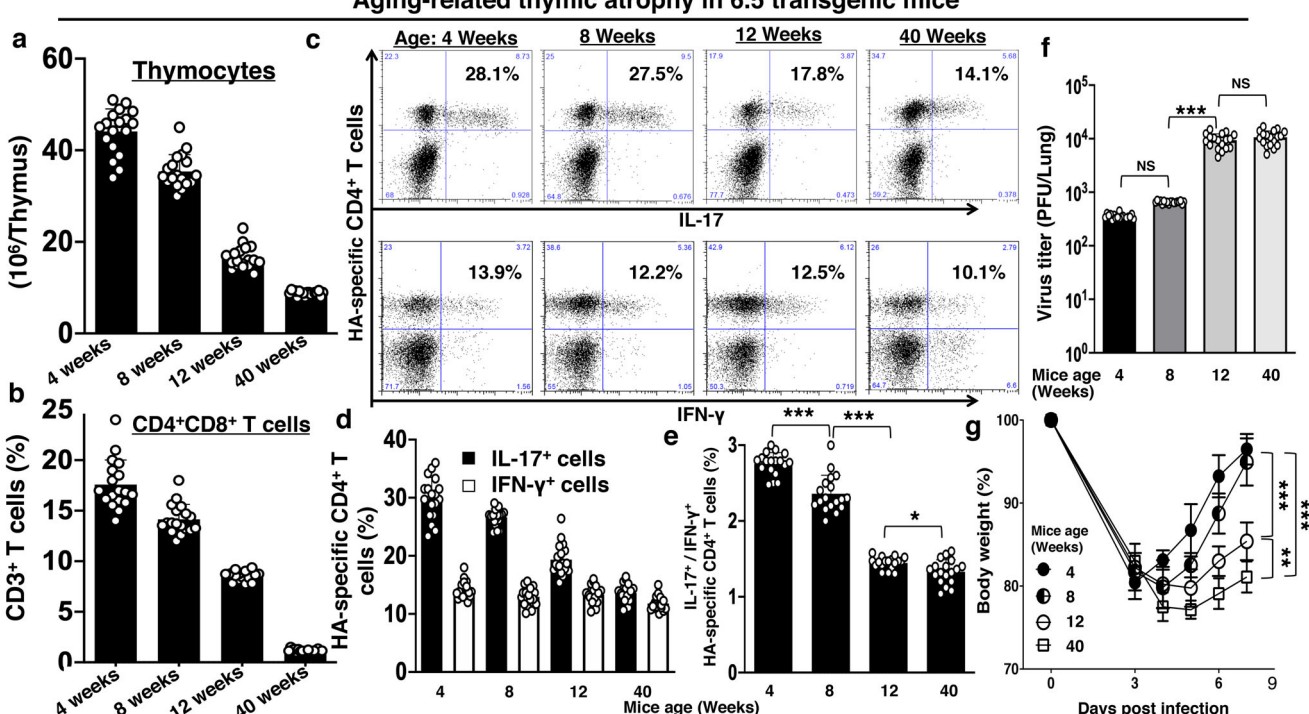

**Fig. 3 Waning Th17 with aging-related thymic involution and severe disease of influenza virus infection in older than younger adult 6.5 mice.**
**a** Thymocyte numbers and **b** CD4[+]CD8[+] cell percentage in CD3[+] thymocytes in healthy non-infected 6.5 mice of stated ages. **c** Dot-plots and **d** bar graphs display waning IL-17 and maintained IFN-γ responses of lung-infiltrating HA-specific CD4[+] T cells with age of 6.5 mice on day 7 after infection. **e** Lost dominance of IL-17 response over the IFN-γ response of lung-infiltrating HA-specific CD4[+] T cells with age of 6.5 mice on day 7 after infection. **f** Higher virus burden in the lungs on day 7 after infection, and **g** delayed recovery of bodyweight loss with age in infected 6.5 mice. Dot-plots are representatives, and other values are means ± s.d. of at least 3 experiments ($n = 6$/group; ***$p < 0.0001$; *$p < 0.01$; NS non-significant, $p > 0.05$; two-tailed $P$ values for unpaired $t$-test to compare two groups at a specific time point, Two-way ANOVA for body weight curves).

cells with IL-17KO donor 6.5 cells abolished the protection with two transfers of IL-17 competent donor 6.5 cells (Fig. 6b). Despite facilitated virus clearance with the donor IL-17KO 6.5 cells (Fig. 6c), the mice suffered from aggravated disease with higher mortality with less IL-17 (Fig. 6d, f) and more inflammatory IFN-γ production in donor 6.5 cells of both transfers (Fig. 6e, f).

**Influenza virus-activated TGF-β facilitates the evolution of Th17 response.** The neuraminidase of the PR8 influenza virus activates TGF-β of the HA-specific CD4[+] T cells[18], and TGF-β is an indispensable cytokine for Th17 polarization in vitro[25]. Naïve HA-specific 6.5 CD4[+] T cells were adoptively transferred into wild-type mice (WT mice + 6.5 cells) and the mice got infection of $2.5 \times 10^3$ p.f.u. of PR8 virus. The lung-infiltrating donor 6.5 cells had activated TGF-β on day 4 after infection in wild-type mice (Fig. 7a). They were retrieved from the recipient mice at that time for co-culture with naïve 6.5 CD4[+] T cells in vitro. Without co-culture, the naïve 6.5 CD4[+] T cells were Th1 cells and 16% of them produced IFN-γ after 48 h stimulation with PR8 virus in vitro. When co-cultured with the day 4 lung-infiltrating donor 6.5 cells, the naïve 6.5 CD4[+] T cells were skewed to Th17 cells. Around 20% of them produced IL-17 and only 7% of them produced IFN-γ (Fig. 7b). Recombinant mouse TGF-β augmented and anti-TGF-β antibody attenuated the IL-17 production of the 6.5 CD4[+] T cells (Fig. 7c). Vaccinia virus carrying HA (Vac-HA) activated adoptively transferred naïve HA-specific 6.5 CD4[+] T cells in vivo to comparable amplitude as PR8 influenza virus. As Vac-HA does not have neuraminidase, donor 6.5 cells did not get activated TGF-β. After retrieval from the recipient mice on day 4,

these Vac-HA activated lung-infiltrating donor 6.5 cells did not skew the Th1 cells into Th17 cells in the in vitro culture (Fig. 7).

**Influenza virus-activated TGF-β of the Th1 cells guide naïve cells to respond with Th17 phenotype.** In our earlier report, we demonstrated augmented TGF-β activation of the HA-specific CD4[+] T cells in the absence of IL-10[18]. Here in, we tried two transfers of IL-10KO donor 6.5 cells into IL-10KO mice (IL-10KO mice + 2-batches of IL-10KO 6.5 cells; IL-10KO set), first transfer with PR8 influenza virus infection and second transfer on 4 days post infection. Two transfers of IL-10-competent donor 6.5 cells into wild-type mice (WT mice + 2-batches of 6.5 cells; WT set) served as controls (Fig. 8a, Supplementary Note 3, Supplementary Fig. 4). There was higher active-TGF-β in the IL-10KO donor 6.5 cells of the first transfer in IL-10KO setting with no IL-10, compared to the IL-10-competent donor 6.5 cells of the first transfer in WT setting with IL-10 (Fig. 8b). In addition, there was higher IL-17 in the IL-10KO donor 6.5 cells of the second transfer in IL-10KO setting with no IL-10, compared to the IL-10-competent donor 6.5 cells of second transfer in the WT setting with IL-10. About 60% of the second transfer cells produced IL-17 in the absence of IL-10 on day 8 post infection, compared to 25% in the setting with IL-10 (Fig. 8c), and the IL-17 production of the second transfer cells was linked to the protection against influenza virus infection in IL-10KO mice (Supplementary Note 3, Supplementary Fig. 4).

We next performed a similar two-transfer experiment in WT setting (WT mice + 2-batches of 6.5 cells) with HA-inserted vaccinia virus (Vac-HA) instead of PR8 influenza virus infection (Fig. 8d). The donor 6.5 cells of both transfers, transferred with infection and transferred on day 4, were activated into Th1

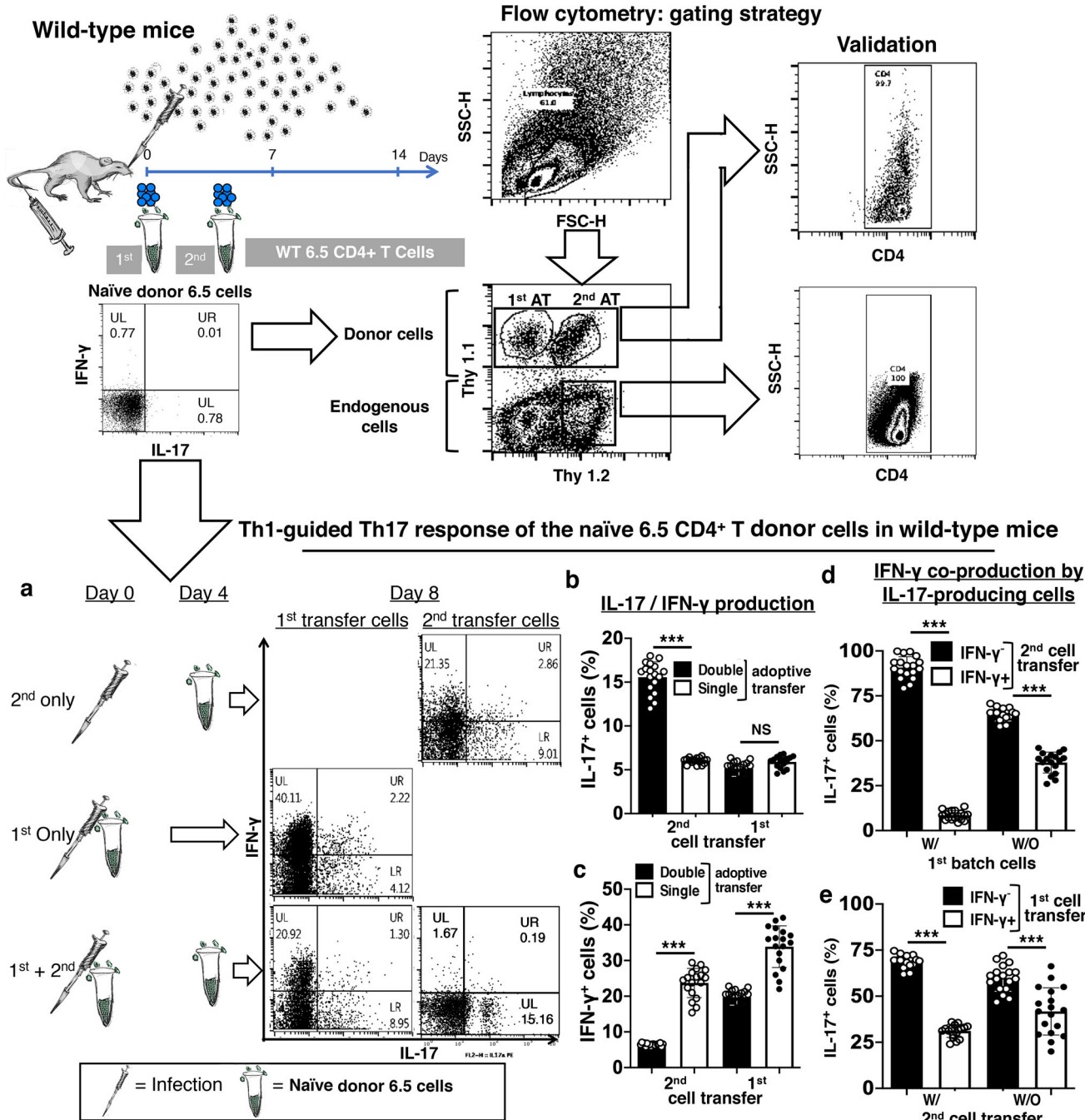

**Fig. 4 Th1-guided Th17 response of naïve cognate antigen-specific CD4+ T cells in the lungs of infected wild-type mice. a** Left panels: Schematic diagram of two-batch naïve HA-specific CD4+ T cell adoptive transfer experiment in infected wild-type mice, as described in text. Right panels: Th17 response of the second-batch cells in the presence of the first-batch cells in the lungs of infected wild-type mice with two-batch cell transfer. **b** IL-17 and **c** IFN-γ production of the first- and second-batch cells on day 8 in the lungs of infected wild-type mice with a single-batch or two-batch HA-specific CD4+ T cell transfer. **d** IFN-γ co-production by the IL-17-producing second-batch HA-specific CD4+ T cells on day 8 in the presence or absence of adoptively transferred first-batch HA-specific CD4+ T cells. **e** IFN-γ co-production by the IL-17-producing first-batch HA-specific CD4+ T cells in the presence or absence of the second-batch cells by this time. Dot-plots are representatives, and other values are means ± s.d. of at least three experiments ($n = 6$/group; ***$p < 0.0001$; *$p < 0.01$; NS non-significant, $p > 0.05$; two-tailed $P$ values for unpaired $t$-test).

phenotype by the Vac-HA infection, as expected with the absence of active-TGF-β on the first transfer donor 6.5 cells (Fig. 8e).

**IL-17 signaling through the non-canonical IL-17 receptor EGFR activates the scaffold protein TRAF4 more than TRAF6 in the lung-infiltrating HA-specific CD4+ T cells.** IL-17 signaling through its well-known receptor IL-17RA contributes to

inflammation[26]. Epidermal Growth Factor Receptor (EGFR; ErbB-1; HER1) has been described as a non-canonical receptor for IL-17. IL-17 signaling with EGFR has been demonstrated to participate in wound healing process[27]. We infected 6.5 mice with $2.5 \times 10^3$ p.f.u. of PR8 strain of influenza virus. Wild-type mice received adoptive transfer of naïve 6.5 donor cells with the infection (WT mice + 6.5 cells; WT set), and IL-17KO

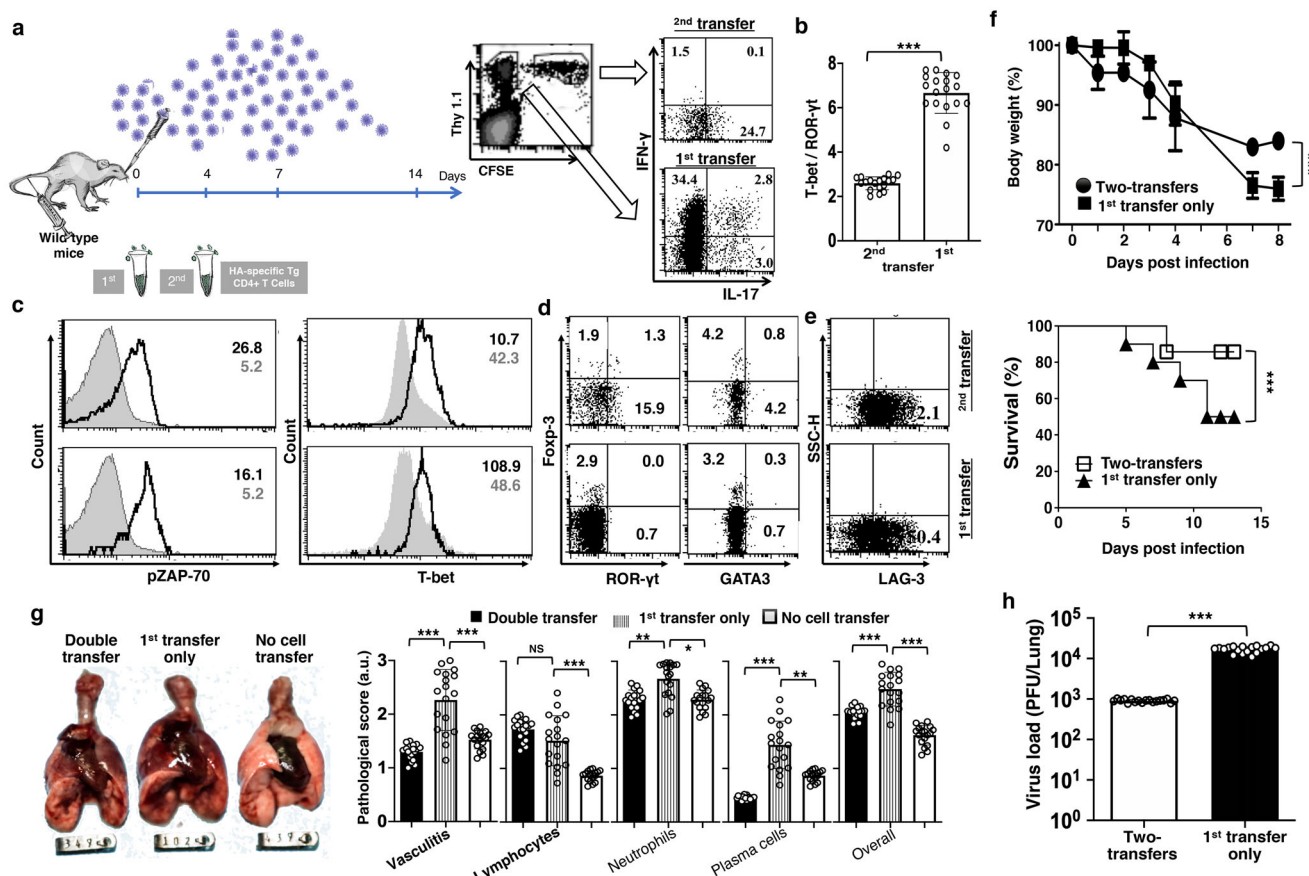

**Fig. 5 Th1-guided Th17 response with reduced dominance of T-bet over ROR-γt alleviates lung inflammation and bestows protection in infected wild-type mice. a** Schematic diagram and Th17 response with **b** reduced dominance of T-bet over ROR-γt of the second-batch donor cells in the two-batch naïve HA-specific CD4+ T cell adoptive transfer experiment in infected wild-type mice. **c** Comparable strength of TCR signaling from ZAP70- phosphorylation to T-bet induction in the first-batch Th1 and second-batch Th17 cells in the lungs on day 8 after infection. Numbers represent MFI (Mean Fluorescence intensity) of the stated molecules. **d** Unaltered Foxp-3+ population size with higher ROR-γt activation of the second-batch compared to the first-batch donor cells on day 8. **e** Higher expression of iTreg marker LAG-3 on the surface of the second-batch compared to the first-batch donor cells by this time. **f** Facilitated recovery of body weight loss and improved survival of infected wild-type mice with two-batches of naïve HA-specific CD4+ T cell adoptive transfer than the mice with a single-batch cognate cell transfer. **g** Gross lung pathology and pathological scores of lung sections, and **h** Virus burden in the lungs of infected wild-type mice with two-batches of naïve HA-specific CD4+ T cell adoptive transfer than the mice with a single-batch cognate cell transfer. Photographs, histograms and dot-plots are representatives, and other values are means ± s.d. of at least three experiments (*n* = 6/group; ***p < 0.0001; **p < 0.001; *p < 0.01; *NS* non-significant, p > 0.05; two-tailed P values for unpaired *t*-test to compare two groups at a specific time point, Two-way ANOVA for body weight curves, and Gehan–Breslow–Wilcoxon test for survival curves).

mice received adoptive transfer of naïve IL-17KO 6.5 donor cells with the infection (IL-17KO mice + IL-17KO 6.5 cells; IL-17KO set). There was up-regulation of both IL-17RA and EGFR on the lung-infiltrating CD4+ T cells in 6.5 mice after the influenza virus infection. IL-17RA was up-regulated only on day 3 post infection, but not thereafter. EGFR was up regulated gradually from day 3 to day 6 and day 9-post infection (Fig. 9a). EGFR was more abundant than IL-17RA on the cell surface from day 6 to day 9 post infection (Fig. 9b). Up-regulation of IL-17RA and EGFR were also there on lung-infiltrating donor 6.5 and IL-17KO 6.5 cells of WT set and IL-17KO set, respectively (Fig. 9a, b, Supplementary Note 4, Supplementary Fig. 5). However, there was EGFR dominance over IL-17RA on lung-infiltrating CD4+ T cell surfaces on day 6 but not on day 9 in WT set. There was no EGFR dominance over IL-17RA on days 6 and 9 in IL-17KO set (Fig. 9b).

Infected 6.5 mice had an abundance of EGFR on the surface of lung-infiltrating CD4+ T cells, and it was associated with higher phosphorylation of EGFR on day 9 than day 6 and day 3. In contrast, only a base level phosphorylated-EGFR was

detected in the lung-infiltrating donor 6.5 cells of WT set and donor IL-17KO cells of IL-17KO set on days 3, 6, and 9 (Fig. 9c). Upon stimulation of splenocytes from naïve 6.5 mice, IL-17KO 6.5 mice, or WT mice with 0.03 MOI influenza virus for 48 h in vitro, there was more EGFR phosphorylation of 6.5 CD4+ T cells than that of IL-17KO 6.5 CD4+ T cells (Fig. 9d). IL-17 supplement augmented EGFR phosphorylation of IL-17KO 6.5 CD4+ T cells in a dose dependent manner, more augmentation with 1.0 than 0.1 µg/ml of recombinant mouse IL-17 (Fig. 9e). EGFR inhibitor Gefitinib mitigated EGFR phosphorylation of 6.5 CD4+ T cells and IL-17-induced EGFR phosphorylation of IL-17KO 6.5 CD4+ T cells (Fig. 9e). Wild-type CD4+ T cells from naïve WT mice served as controls, in which, IL-17 supplement augmented and Gefitinib attenuated EGFR phosphorylation (Fig. 9e).

Tumor Necrosis Factor Receptor-Associated Factors (TRAF) family (TRAF 1 to 7) of seven scaffold proteins are involved in various cellular physiological processes[28]. TRAFs, as adapter molecules, moderate many signaling pathways. TRAF4 and TRAF6 share the same TRAF binding sites, but they play

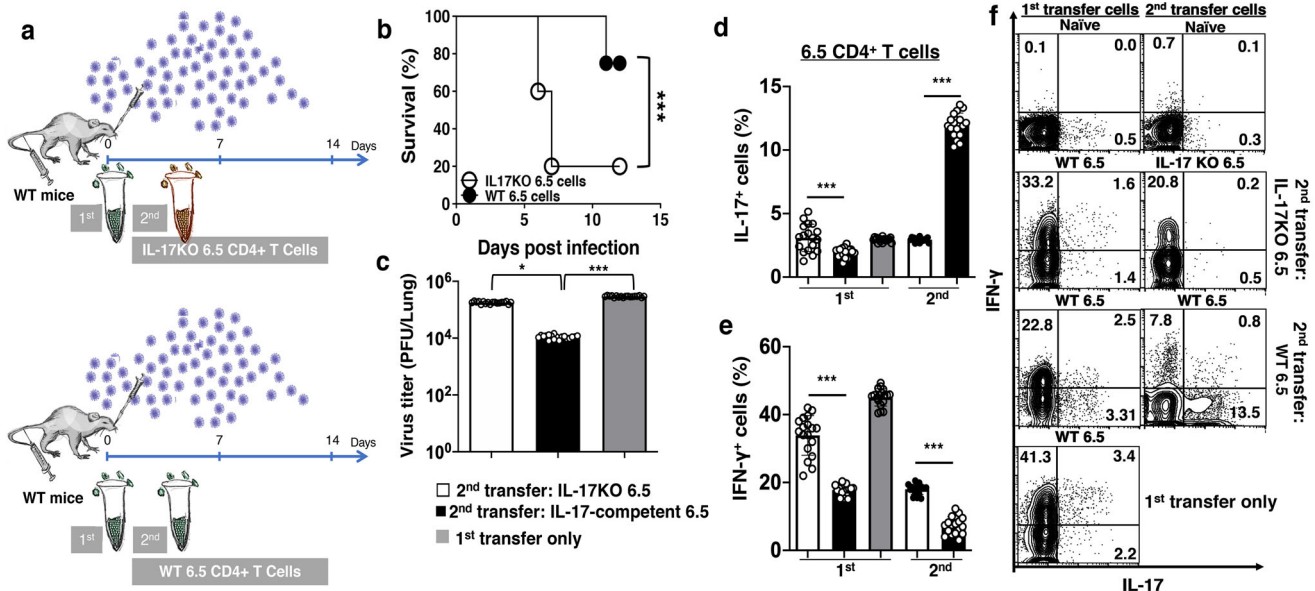

**Fig. 6 Lost protection of two-batch adoptive transfer experiment with the use of naïve IL-17KO HA-specific CD4+ T cells in the second-batch cell. a** Schematic diagram and **b** lost protection of two-batch adoptive transfer experiment in infected wild-type mice with the adoptive transfer of naïve IL-17KO HA-specific CD4+ T cells in the second batch. **c** Day 8 virus burden in the lungs of infected wild-type mice in two-batch naïve HA-specific CD4+ T cell transfer experiment, with IL-17KO or IL-17 competent second-batch cells. **d** Unaltered IL-17 production by the first-batch cells, and **e** boosted IFN-γ production by both first- and second-batch cells with the second-batch naïve IL-17KO HA-specific CD4+ T cell transfer. **f** Representative dot-plots of IL-17 and IFN-γ production by both first- and second-batch cells with the second-batch naïve IL-17-competent or IL-17KO HA-specific CD4+ T cell transfer. Values are means ± s.d. of at least three experiments (n = 6/group; ***p < 0.0001; *p < 0.01; NS non-significant, p > 0.05; two-tailed P values for unpaired t-test to compare two groups at a specific time point, and Gehan–Breslow–Wilcoxon test for survival curves).

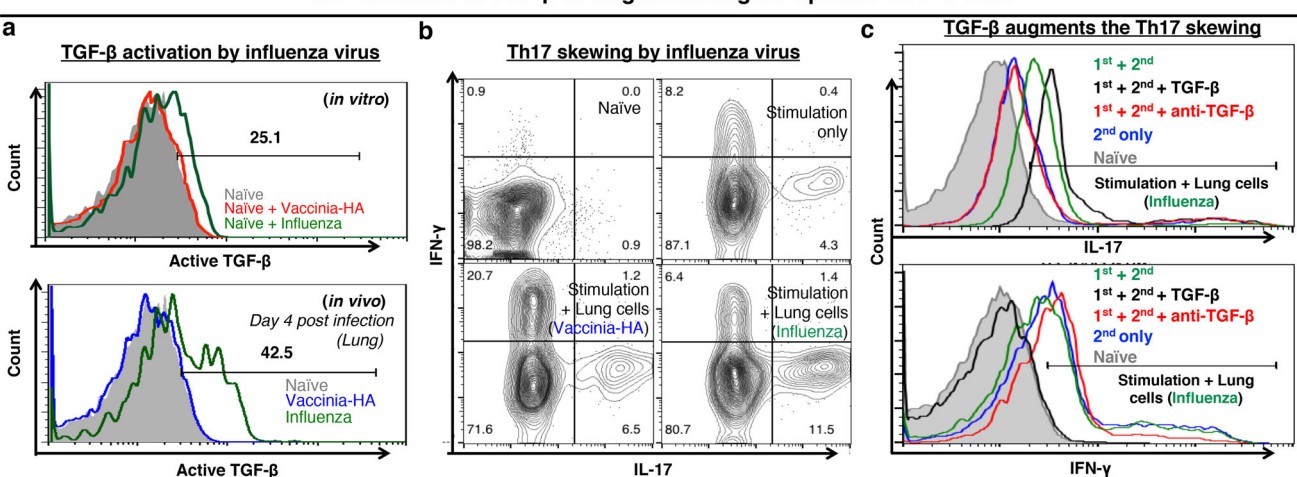

**Fig. 7 Lung cells from influenza virus infected mice guides Th17 response of naïve HA-specific CD4+ T cells, and the response is facilitated by additional TGF-β. a** TGF-β activation in the HA-specific CD4+ T cells by influenza virus, but not by HA-inserted vaccinia (Vac-HA) virus. Upper panel: Splenocytes from HA-specific TCR-transgenic 6.5 mice were stimulated overnight with influenza or Vac-HA virus. Lower Panel: Wild-type mice received naïve HA-specific CD4+ T cell adoptive transfer with influenza or Vac-HA virus infection, and lung cells were analyzed on day 4. **b** The Day 4 lung cells were mixed with naïve splenocytes from HA-specific TCR-transgenic 6.5 mice and stimulated with influenza virus for 48 h in vitro. **c** Addition of recombinant mouse TGF-β boosted, and anti-mouse TGF-β antibody attenuated IL-17 production by the HA-specific CD4+ T cells in the co-culture. Histograms and dot-plots are representatives.

different roles in the pathogenesis[29,30]. A real-time quantitative reverse transcription PCR study of Day 6 lung cells revealed up-regulation of TRAF4 and TRAF6 in the infected 6.5 mice. The TRAF4 was dominant over TRAF6 in the lung-infiltrating CD4+ T cells of infected 6.5 mice (Fig. 9f). In contrast, TRAF6 was dominant over TRAF4 in the lung-infiltrating donor 6.5 CD4+ T cells of infected wild-type (WT mice + 6.5 cells; WT set)

or donor IL-17KO 6.5 CD4+ T cells of infected IL-17KO mice (IL-17KO mice + IL-17KO 6.5 cells; IL-17KO set; Fig. 9f). Upon similar in vitro stimulation, as described above, flow cytometry revealed more up-regulation of TRAF4 protein in 6.5 than IL-17KO 6.5 CD4+ T cells (Fig. 9g). IL-17 supplement augmented TRAF4 up-regulation in IL-17KO 6.5 CD4+ T cells in a dose dependent manner, more augmentation with 1.0 than 0.1 µg/ml

**Facilitated Th17 evolution w/ augmented TGF-β of first transfer Th1 cells in acute influenza in the absence of IL-10**

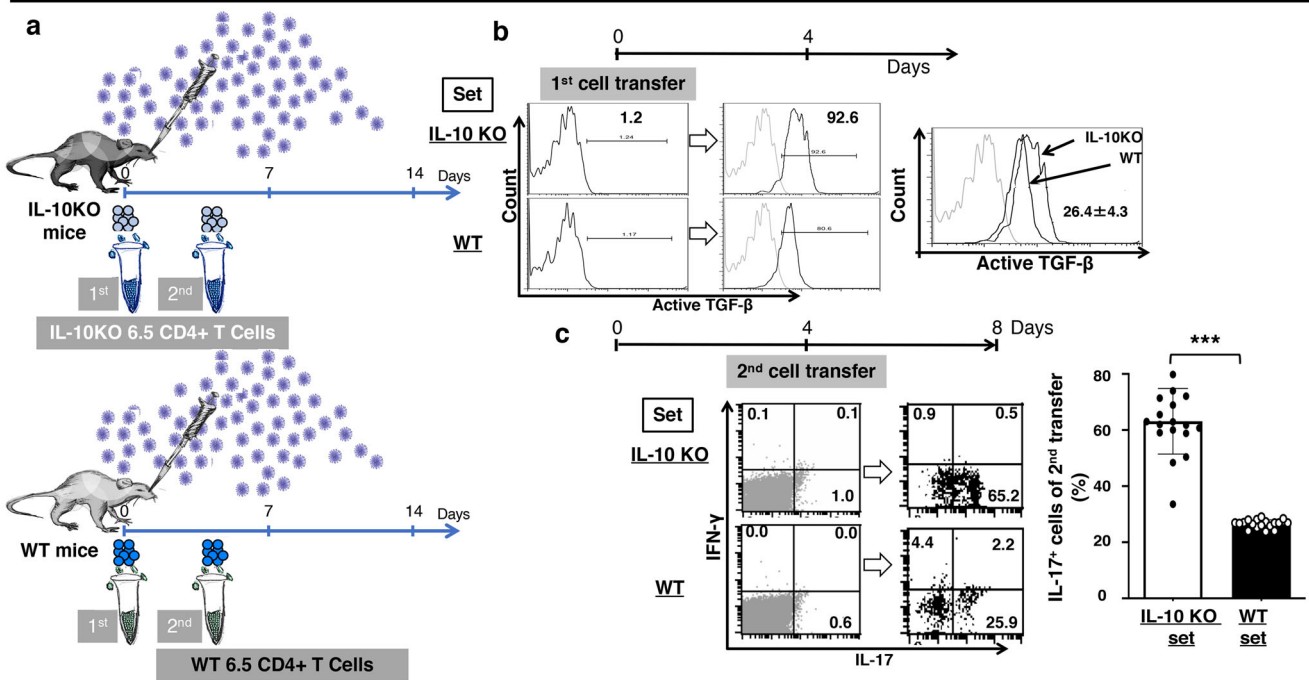

**Lack of Th17 evolution W/O TGF-β activation by vaccinia virus infection**

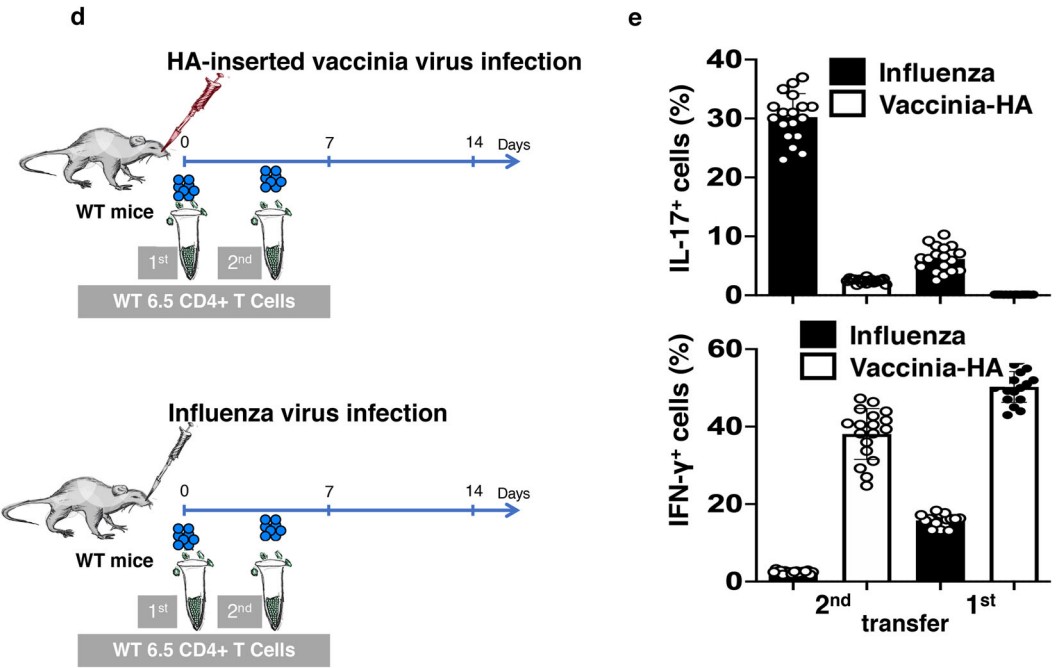

**Fig. 8 Influenza virus-activated TGF-β of the Th1 cells guides Th17 response of naïve cognate antigen-specific CD4$^+$ T cells. a–c** Two-batch adoptive transfer experiment in the absence of IL-10. **a** Schematic diagram: naïve IL-10KO HA-specific CD4$^+$ T cells were adoptively transferred into IL-10KO mice on days 0 and 4 after infection (IL-10KO set). Infected WT mice with IL-10 competent HA-specific CD4$^+$ T cells (WT set) served as control. **b** More TGF-β of first-batch IL-10KO donor cells and **c** intensified Th17 response of the second-batch IL-10KO donor cells in IL-10KO set on day 8. **d, e** Two-batch adoptive transfer experiment in wild-type mice with HA-inserted vaccinia (Vac-HA) virus infection. **d** No Th17 response of the second-batch donor cells in mice with Vac-HA infection. Histograms and dot-plots are representatives, and other values are means ± s.d. of at least three experiments ($n = 6$/group; ***$p < 0.0001$; two-tailed $P$ values for unpaired $t$-test).

of recombinant mouse IL-17 (Fig. 9g). EGFR inhibitor Gefitinib mitigated TRAF4 up-regulation in 6.5 CD4$^+$ T cells and IL-17-induced TRAF4 up-regulation in IL-17KO 6.5 CD4$^+$ T cells (Fig. 9g). Wild-type CD4$^+$ T cells from naïve WT mice served as

controls, in which, IL-17 supplement augmented and Gefitinib attenuated TRAF4 up-regulation (Fig. 9g).

Confocal microscopy revealed TRAF4 translocation to EGFR signaling complex in the 6.5 CD4$^+$ T cells, upon stimulation of

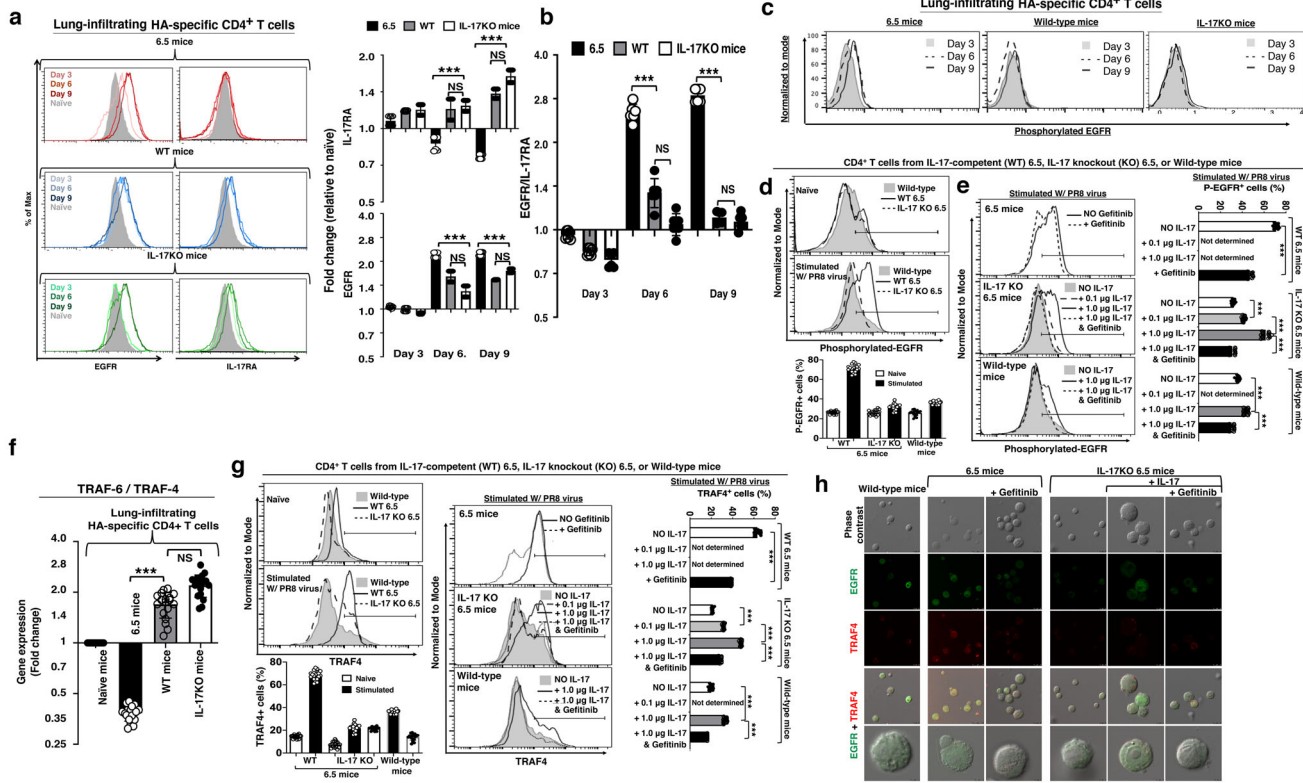

**Fig. 9 The higher activation of TRAF4 than TRAF6 is associated with the abundance of EGFR over IL-17RA in influenza virus infected 6.5 mice. a** EGFR and IL-17RA expressions on the surface of lung-infiltrating CD4+ T cells in infected TCR-transgenic 6.5 mice, wild-type mice and IL-17KO mice. Non-infected healthy mice (naïve) served as controls. Bar graphs display Up-or down-regulation of EGFR and IL-17RA expressions with time after infection in these mice. **b** Abundance of EGFR than IL-17RA on the surface of lung-infiltrating CD4+ T cells on days 6 and 9 after infection in 6.5 mice. **c** Higher EGFR phosphorylation of lung-infiltrating CD4+ T cells in the infected 6.5 mice on days 6 and 9. **d–e** EGFR phosphorylation in the 6.5 CD4+ T cells after 48 h of stimulation with 0.03 MOI PR8 virus in vitro. Splenocytes from naïve IL-17-competent and IL-17KO 6.5 mice were stimulated, and splenocytes from wild-type mice served as controls. Mouse recombinant IL-17 (0.1 or 1.0 μg/ml) or Gefitinib (1.0 μM) was added to the culture condition, as stated in the text. **f** Dominance of TRAF4 over TRAF6 in the lungs of infected 6.5 mice, in contrast to the dominance of TRAF6 over TRAF4 in the lungs of infected wild-type, and IL-17KO mice as determined by mRNA levels in lung-infiltrating CD4+ T cells and described in text. **g** Intracellular staining of TRAF4 in the 6.5 CD4+ T cells after 48 h of stimulation with 0.03 MOI PR8 virus in vitro. Splenocytes from naïve IL-17-competent and IL-17KO 6.5 mice were stimulated, and splenocytes from wild-type mice served as controls. Mouse recombinant IL-17 (0.1 or 1.0 μg/ml) or Gefitinib (1.0 μM) was added to the culture condition, as stated in the text. **h** Co-localization of EGFR and TRAF4 in the 6.5, IL-17KO 6.5, and wild-type CD4+ T cells after 48 h of stimulation with 0.03 MOI PR8 virus in vitro. Histograms and photographs are representatives, and other values are means ± s.d. of at least three experiments. ($n = 6$/group; ***$p < 0.0001$; NS non-significant, $p > 0.05$; two-tailed $P$ values for unpaired $t$-test).

splenocytes from naïve 6.5 mice with 0.03 MOI influenza virus for 48 h in vitro (Fig. 9h). The co-localization of TRAF4 and EGFR was more significant in the 6.5 CD4+ T cells than that of IL-17KO 6.5 CD4+ T cells. IL-17 supplement augmented TRAF4 and EGFR co-localization in the stimulated IL-17KO 6.5 CD4+ T cells (Fig. 9h). EGFR inhibitor Gefitinib mitigated TRAF4 and EGFR co-localization in 6.5 CD4+ T cells and IL-17-induced TRAF4 and EGFR co-localization in IL-17KO 6.5 CD4+ T cells (Fig. 9h).

**Treatment with EGFR inhibitor, Gefitinib, aggravates the disease of severe influenza.** Gefitinib is an EGFR inhibitor, which interrupts signaling through the epidermal growth factor receptor (EGFR) in target cells. Gefitinib aggravated the disease of influenza virus infection in mice. The impact of EGFR inhibition with Gefitinib was highest in infected 6.5 mice, moderate in infected wild-type mice, and negligible in infected IL-17KO mice (Fig. 10a). In the infected wild-type mice, intranasal IL-17 supplement bestowed some protection, and the protection was lost with the treatment of Gefitinib (Fig. 10b). The lost benefit was associated with altered up-regulation of TRAF4 and TRAF6 in

the lungs. The IL-17 supplement altered the TRAF6 dominance over TRAF4 in the lungs of infected wild type mice, and the alteration was restricted when the IL-17 supplement was given with Gefitinib treatment (Fig. 10b).

**Discussion**

Here we demonstrated that influenza virus-activated TGF-β of the Th1 cells guides the Th17 evolution of recent thymic emigrants. IL-17 signaling through the non-canonical IL-17 receptor EGFR activates the scaffold protein TRAF4 more than TRAF6 during alleviation of lung inflammation in severe influenza. Adoptively transferred HA-specific 6.5 CD4+ T cells caused collateral damage with their IFN-γ-producing Th1 response to the infection in wild-type mice. The donor mice of the naïve HA-specific 6.5 CD4+ T cells, the 6.5 TCR transgenic mice, have all the CD4+ T cells responsive to the HA antigen of the influenza virus. The overwhelming CD4+ T cell response in the infected 6.5 mice would cause more damage and disease of more severity. There was aggravated illness at first, but the subsequent evolution of a prominent Th17 response mitigated the disease and the 6.5 mice had better survival than wild-type mice. Both Th1 and Th17

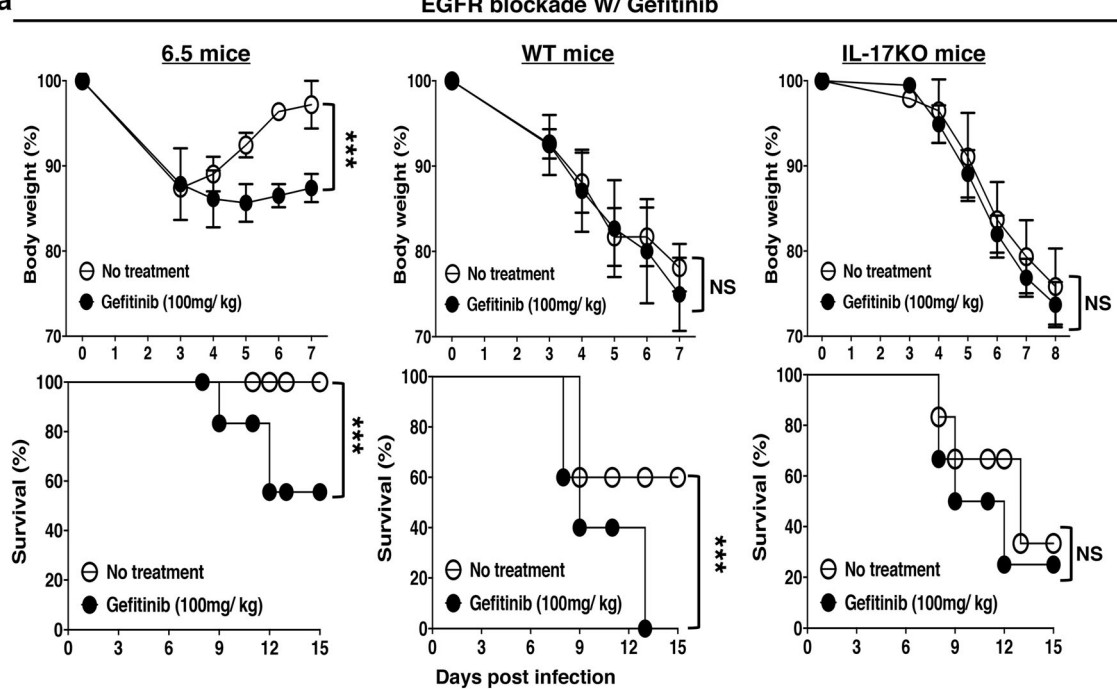

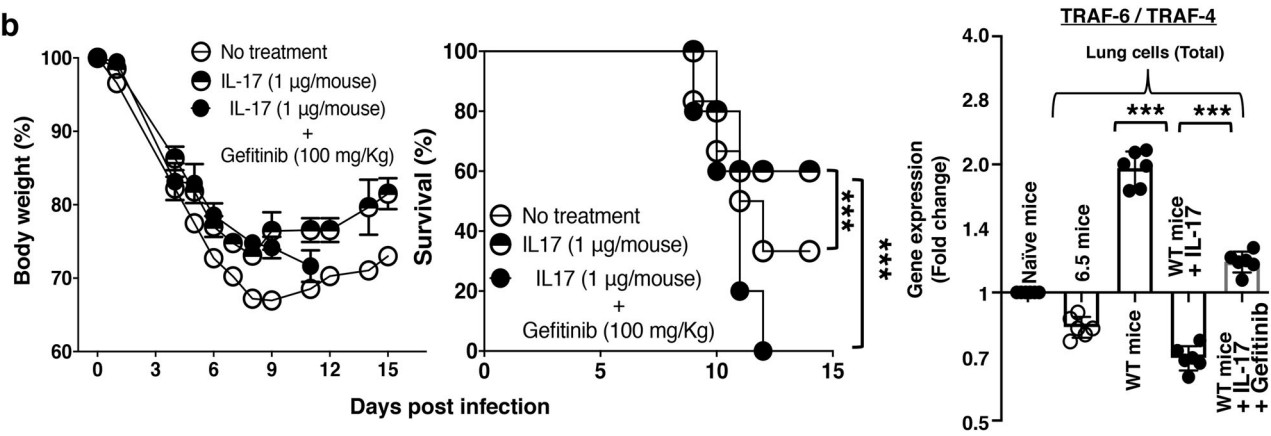

**Fig. 10 Disrupted IL-17-EGFR axis with Gefitinib treatment alters TRAF4 dominance and disease severity in acute influenza. a** EGFR inhibition with Gefitinib resulted in more body weight loss and higher mortality of influenza virus infected 6.5 mice. Infected wild-type and IL-17KO mice served as controls. **b** Gefitinib treatment also diminished the protection conferred by IL-17 supplement, and resulted in more body weight loss and higher mortality of infected wild-type mice. The alterations were associated with altered TRAF4 dominance over TRAF6 mRNA levels. Values are means ± s.d. of at least three experiments. ($n = 6$/group; ***$p < 0.0001$; NS non-significant, $p > 0.05$; two-tailed $P$ values for unpaired $t$-test to compare two groups at a specific time point, Two-way ANOVA for body weight curves, and Gehan–Breslow–Wilcoxon test for survival curves).

CD4+ T cells were activated for Th1 commitment with comparable TCR signaling from ZAP-70 phosphorylation to T-bet induction. The Th17 evolution was associated with ROR-γt upregulation and reduced T-bet dominance over ROR-γt. Thymectomy, aged mice experiments, and repetitive adoptive transfer experiments revealed the Th1 CD4+ T cells guided Th17 evolution, associated with viral neuraminidase activated TGF-β on the Th1 cells. IL-17 signaling through the EGFR happened with relative abundance of EGFR to IL-17RA in 6.5 CD4+ TCR-transgenic mice. The signaling activates the scaffold protein TRAF4 more than TRAF6. Gefitinib blockade documented the significance of IL-17 signaling through EGFR in alleviation of lung inflammation in severe influenza.

Both IFNγ and IL-17 are fundamental constituents of the cytokine storm in severe influenza with pneumonia[31]. IFN-γ is a hallmark Th1 cytokine essential for viral clearance[32]. In our experimental system, lung-infiltrating HA-specific CD4+ T cells produce IFN-γ in response to the infection[17,18]. The transcription factor T-bet directs commitment to the Th1 lineage and IFN-γ production[33,34]. Here we found our Th17 evolution was also preceded by Th1 commitment with a comparable strength of TCR signaling from ZAP-70 phosphorylation to T-bet induction. TGF-β is indispensable for Th17 polarization of naïve CD4+ T cells in vitro[25]. We revealed viral neuraminidase-activated TGF-β of the Th1 CD4+ T cells is essential for the Th1 guided evolution of Th17 cells. The Th1 CD4+ T cells steered activation of Th17-specific transcription factor ROR-γt and production of Th17-specific cytokine IL-17.

The Th1 transcription factor T-bet has been shown to suppress Th2 and Th17 differentiation[35,36]. However, co-production of

IFN-γ and IL-17 has been detected in human CD4[+] T cells[25,37] and co-production IFN-γ and IL-10 has been documented in murine CD4[+] T cells[18]. Relative expression levels of the transcription factors T-bet, GATA-3, and ROR-γt may fine tune a balance among different lineages of the conventional Th1/Th2/Th17 paradigm. A diminished T-bet dominance over GATA-3 and ROR-γt obliterates IFN-γ production of lung-infiltrating HA-specific CD4[+] T cells in our experimental system[17]. Here, in the same experimental system of severe influenza, we demonstrated Th17 differentiation with diminished dominance of T-bet over ROR-γt. This excessive ROR-γt activation drove the Th1 committed CD4[+] T cells into the Th17 lineage and abolish the IFN-γ production.

The role IL-17 remained uncertain. IL-17 is linked to localized inflammation and pathogenic in certain autoimmune diseases, in which IL-17 binds to its canonical receptor IL-17RA and activates TRAF6 and the subsequent signaling cascade[26,28,38]. IL-17 is anti-inflammatory in other diseases[39–41]. IL-17 is highly versatile and has been implicated in other non-pathogenic conditions including tissue repair and cancer as well[42], in which IL-17 binds to recruited EGFR in the receptor complex and activates TRAF4 and subsequent signaling cascade[27,39]. TRAF4 activation may restrict IL-17-mediated pathogenic processes[30]. Our HA-specific Th17 cells were similar to the non-pathogenic Th17 cells[25], as they do not produce IFN-γ and do not boost inflammation. There was also signaling through EGFR and subsequent activation of TRAF4. The Th17 response alleviated inflammation in our mouse model of severe influenza.

Aged people are at higher risk of severe influenza[43]. They also suffered from excessive mortality during SARS and the ongoing COVID-19 pandemics[44,45], https://www.cdc.gov/coronavirus/2019-ncov/need-extra-precautions/index.html; accessed August 23, 2022. Their comorbidities contribute to this fact but it may also involve the derangement of the immune system. Age related thymic involution and decreased thymic output may be the problem. At 50 years of age, the human thymus generates only 15–20% of the T cells compared to the young. It is almost unable to make new T cells by the age of 65 (Ref: [46]). Thymic involution starts from three months of age in mice[47]. The disease alleviating Th17 cells displayed the surface markers of recent thymic emigrants. Thymectomy abolished the Th17 response and the Th17 waned with age. The aged people may suffer from severe disease because of loss of the disease modifying Th17 cells. Our results raise alarms regarding modulating IL-17 response in severe respiratory tract infections. We have to be cautious to avoid inappropriate cytokine manipulation as IL-17 is not necessarily always proinflammatory.

## Methods

**Mice and influenza virus infection**. All experiments involving mice were performed in accordance with protocols approved by the "Animal Care and Use Committee" of Chang Gung Memorial Hospital (Approval number: IACUC 2016030301). The committee certified that all experiments involving mice were carried out in strict accordance with the Animal Protection Law by the Council of Agriculture, Executive Yuan, Taiwan, R.O.C., and the guidelines contained in the Guide for the Care and Use of Laboratory Animals, as promulgated by the Institute of Laboratory Animal Resources, National Research Council, U.S.A.

All mice were of C57BL/10.D2 genetic background, and 8–12-week-old mice of both sexes, except in experiments with aged mice, were used for experiments. The TCR-transgenic mouse line 6.5 expresses a TCR recognizing an I-E$^d$-restricted HA epitope ([110]SFERFEIFPKE[120]) of HA of PR8 influenza virus (generously provided by H. Von Boehmer, Harvard University, Boston, MA). The 6.5 mice were crossed with syngeneic IL17KO or IL10KO mice to generate 6.5 IL17KO TCR-transgenic, or 6.5IL10 KO TCR-transgenic mice, respectively. Syngeneic IL17KO mice were generated by breeding IL17KO mice of B6 genetic background (generously provided by Y-C. Day, Chang Gung Memorial Hospital, Taiwan) into mice with a C57BL/10.D2 genetic background for >9 generations. Syngeneic IL10 KO and 6.5 IL10 KO TCR-transgenic mice were described earlier[18]. Mice were maintained under specific pathogen-free conditions.

The PR8 (A/PR/8/34) strain of influenza virus was produced in the allantoic fluid of 10-day-old embryonated chicken eggs, and characterized by a core facility at Chang Gung University, Taiwan. Mice were inoculated via the intranasal route during light isoflurane (Aesica Queenborough Ltd., Kent, UK) anesthesia with $2.5 \times 10^3$ plaque-forming units (p.f.u.s) in 50 μl of PBS[17–22,34].

**Infection with recombinant vaccinia virus encoding HA of PR8 strain of H1N1 influenza virus**. The HA-inserted vaccinia virus[48–50] was amplified using BCS-1 cells. An infective dose of $1 \times 10^6$ plaque-forming units (p.f.u.s) was injected intraperitoneally in 100 μl of PBS.

**IL-17 supplement & EGFR blockade**. Recombinant mouse IL-17A (carrier free) was purchased from BioLegend (San Diego, CA, USA; Cat# 576006). An appropriate concentration of IL-17A in 20 μl PBS was supplemented daily through intranasal route for 5 days during isoflurane (Aesica Queenborough Ltd., Kent, UK) anesthesia.

The EGFR inhibitor Gefitinib was purchased from AstraZeneca. A stock solution in DMSO was first made and aliquots were kept in −80 °C. A fresh final solution was prepared every day in PBS and 100 mg/kg Gefitinib in 100 μl was fed by oral gavage.

**Adoptive transfer and CFSE staining**. Clonotypic HA-specific CD4[+] TCR transgenic T cells were prepared from pooled spleens and lymph nodes of HA-specific CD4[+] TCR-transgenic 6.5 mice[17–22,34,48,49]. Clonotypic IL17 KO or IL10 KO HA-specific CD4[+] TCR-transgenic T cells were prepared from pooled spleens and lymph nodes of 6.5 IL17 KO or 6.5 IL10 KO TCR-transgenic mice, respectively. Clonotypic percentages were determined by flow cytometric analysis. Naïve phenotype was confirmed by assessing profiles of the activation markers CD44 (anti-mouse CD44; 553133, BD Biosciences; dilution = 1 in 100) and CD62L (anti-mouse CD62L; 553150, BD Biosciences; dilution = 1 in 100). Infection-responding 6.5 CD4[+] T cells were retrieved from the lungs of infected HA-specific CD4[+] TCR-transgenic 6.5 mice. CD4[+] T cells were enriched by negative selection using magnetic beads, and donor cells were identified using fluorochrome-conjugated antibodies against CD90.1 (Thy1.1; anti-mouse CD90.1; 554898, BD Biosciences; dilution = 1 in 1000) or CD90.2 (Thy1.2; anti-mouse CD90.2; 553006, BD Biosciences; dilution = 1 in 200). For single-batch transfer experiments, $2.5 \times 10^6$ clonotypic 6.5 CD4[+] T cells were injected via tail vein into Thy1.2/Thy1.2[+] wild-type recipient mice in 0.2 ml of Hank's Balanced Salt Solution (HBSS). For two-batch transfer experiments, a first batch of $0.5 \times 10^6$ clonotypic 6.5 CD4[+] T cells was injected at the time of infection, and a second batch of $1.5 \times 10^6$ clonotypic 6.5 CD4[+] T cells was injected on day 4 of infection in Thy1.2/Thy1.2[+] wild-type mice.

Clonotypic T cells ($1 \times 10^7$ cells/ml) were stained with 5 μM CFSE (5,6-carboxyfluorescein diacetatesuccinimidyl ester; Molecular Probes, Eugene, OR, USA) in HBSS for 10 min. Cells were washed three times with HBSS, and CFSE-stained 6.5 CD4[+] T cells ($2.5 \times 10^6$) were transferred via tail vein into recipient mice.

**Flow cytometry**. Cell suspensions were incubated on ice with saturated concentrations of fluorochrome-labeled mAbs in FACS buffer (PBS plus 0.5% BSA and 0.02% NaN$_3$) as recommended by the manufacturers. Donor CD4[+] T cells were identified using the following mAbs: biotin-conjugated anti-clonotypic 6.5 TCR (provided by H. Von Boehmer; dilution = 1 in 100), avidin-PE (A1204, Caltag, Burlingame, CA; dilution = 1 in 200), or Streptavidin-APC (SA 1005, Caltag, Burlingame, CA; dilution = 1 in 200), and PerCP or FITC-conjugated anti-CD4 (553052 or 553047, BD Biosciences; dilutions = 1 in 100 for both). All fluorochrome-conjugated antibodies were from BD Biosciences, except for T-bet, and mouse IgG1κ isotype control (45–5825, 45–4714, eBiosciences; dilution = 1 in 50 for both), FITC-Phospho-EGFR (Tyr1068) Rabbit anti-Human, Mouse (MA5-27995, Invitrogen; dilution = 1 in 10), and PE-TRAF4 (SC-390232, Santa Cruz Biotechnology; no dilution). Cells were acquired using FACS CALIBUR, and analysis was performed using CellQuest Pro (BD Biosciences), FACSuite (BD Biosciences), or FlowJo (Tree Star, Inc.) software.

**Intracellular cytokine staining**. Single cell suspensions ($5–10 \times 10^6$ cells well$^{-1}$ in 24 well plates) from harvested organs were re-stimulated for 5 h with 10 μg ml$^{-1}$ MHC class II restricted HA peptide (SFERFEIFPKE) in the presence of 5 μg ml$^{-1}$ Brefeldin A (SI-B7651, Sigma-Aldrich). The Brefeldin-A concentration was maintained throughout intracellular cytokine staining. Re-stimulated cells were surface-stained with anti-CD90.1 and anti-CD4 as described above, fixed in IC fixation buffer (00–8222, eBiosciences), washed, and stained in permeabilization buffer (00–8333, eBiosciences) containing fluorochrome-conjugated antibodies against target cytokines. For Foxp3 (11–5773, eBiosciences; dilution = 1 in 50) and T-bet, cells were surface-stained, fixed, and permeabilized in fixation/permeabilization buffer (00-5523, eBiosciences), washed, and stained in permeabilization buffer (00-8333, eBiosciences) containing fluorochrome-conjugated antibodies. In some experiments, cells were analyzed ex vivo and stained without re-stimulation.

**In vitro proliferation of naïve HA-specific CD4[+] T cells**. Single cell suspensions were prepared from the spleens of HA-specific TRC transgenic 6.5 mice. Splenocytes ($2.5 \times 10^5$) were re-stimulated for 72 h with class-II restricted HA peptide

(SFERFEIFPKE) at 37 °C in the presence of 5% $CO_2$. Tritiated thymidine was added 18 h prior to harvest. The amount of radioactivity incorporated into DNA in each well was measured in a scintillation counter, as described earlier[17,34]. We co-cultured lung-infiltrating HA-specific $CD4^+$ T cells retrieved from infected mice as described in the text, to measure their impact on proliferation.

**Cytokine production and expression of TRAF4 and p-EGFR following in vitro stimulation.** For cytokine production, single-cell suspensions of splenocytes ($2.5 \times 10^6$ cells per well) from IL-17-competent (IL-17$^{+/+}$) 6.5 mice were stimulated with influenza virus (PR8 strain; 0.03 (MOI) Multiplicity of Infection) in complete RPMI-1640 for 48 h in vitro, in the presence of lung-infiltrating HA-specific 6.5 CD4+ T cells from day 4 infected mice at 1:5 ratio. Naïve splenocytes without cells from the infected mice served as controls. Recombinant mouse TGF-β (10 ng ml$^{-1}$, GF346, Millipore) or anti- TGF-β (10 μgml$^{-1}$, MAB1835, R&D) was added to the culture at the onset. Cognate HA peptide (100 μgml$^{-1}$) was added to the culture for last 5 h. Brefeldin-A was added for last 3 h of stimulation and cells were then processed for intracellular staining.

For TRAF4 staining, single-cell suspensions of splenocytes ($2.5 \times 10^6$ cells per well) from naive WT, IL-17-competent (IL-17$^{+/+}$) 6.5, or IL-17 knockout (IL-17$^{-/-}$) 6.5 mice were stimulated with influenza virus (PR8 strain; 0.03 (MOI) Multiplicity of Infection) in complete RPMI-1640 for 48 h in vitro. Recombinant mouse IL-17 (0.1 or 1.0 μgml$^{-1}$; 576006, Biolegend) and/or Gefitinib (0.1 or 1.0 μM; Iressa, AstraZeneca) was added to the culture at the onset. Cognate HA peptide (100 μgml$^{-1}$) was added to the culture for last 5 h. Brefeldin-A was added for last 3 h of stimulation and cells were then processed similarly, as we did for intracellular staining.

For phosphorylated-EGFR staining, single-cell suspensions of splenocytes ($2.5 \times 10^6$ cells per well) from naive WT, IL-17-competent (IL-17$^{+/+}$) 6.5, or IL-17 knockout (IL-17$^{-/-}$) 6.5 mice were stimulated with influenza virus (PR8 strain; 0.03 (MOI) Multiplicity of Infection) in complete RPMI-1640 for 48 h in vitro. Recombinant mouse IL-17 (0.1 or 1.0 μgml$^{-1}$; 576006, Biolegend) and/or Gefitinib (0.1 or 1.0 μM; Iressa, AstraZeneca) was added to the culture at the onset. Cognate HA peptide (100 μgml$^{-1}$) was added to the culture for last 5 h. Brefeldin-A was added for last 3 h of stimulation and cells were then processed for staining. In brief, cells were first placed in cold perm buffer-III (558050, BD Biosciences) for 30 min, washed with stain buffer (554657, BD Biosciences), and stained with FITC-Phospho-EGFR (Tyr1068) Rabbit anti-Human, Mouse (MA5-27995, Invitrogen; dilution = 1 in 10) in stain buffer. Cells were acquired with BD LSRFortessa™ Cell Analyzer for flow cytometry or visualized in confocal microscopy (Leica).

**Plaque assay for virus titer.** We measured live influenza virus titers in the organs of infected mice using a modified Madin-Darby canine kidney cell (MDCK; The American Type Culture Collection) plaque assay, as described earlier[17–22,34]. The organs were collected at the indicated times in 1 mL DMEM, snap-frozen in liquid nitrogen, and stored at −80 °C until analysis. Ten-fold dilutions of tissue homogenates were prepared in DMEM supplemented with 10% FCS, antibiotic-antimycotic (15240-062, GIBCO BRL), and 0.00025% trypsin. A total of 500 μl of each dilution was added to confluent monolayers of MDCK cells in 6 well plates and incubated for 1 h at 35 °C with 5% $CO_2$. Each well then received 2 mL of agar overlay (0.3%) and was incubated for 3 days. Cells were fixed with 10% formalin, agar overlay was removed, and fixed monolayers were stained with crystal violet (0.02% in 2% ethanol). The results were presented as plaque forming unit (PFU)/ ml = (mean number of plaques × 2) × (dilution factor$^{-1}$).

**Histopathology.** The lungs from experimental mice were harvested on day 7 post-infection, and fixed with 10% neutral buffered formalin solution. Following fixation, lungs were embedded in paraffin and 5 μm sections were cut. Sections were stained with hematoxylin and eosin (H&E) and scored blindly. Infiltration of inflammatory cells, including lymphocytes, neutrophils, and plasma cells, was separately scored by pathologists as negative, $1^+$, $2^+$, or $3^+$ according to the density of infiltration. Vasculitis and fibrosis were also scored as negative, $1^+$, $2^+$, or $3^+$ based on severity, as described earlier[17–22,34]. Overall inflammation in the lungs was represented by the average of these scores.

**Statistics and reproducibility.** Data are expressed as mean ± s.d. We used Graphpad Prism version 8 for Student's $t$ test analyses to compare two groups at a specific time point. We also analyzed Two-way ANOVA for the body weight curves and Gehan–Breslow–Wilcoxon test for survival curves. We considered $p$ values < 0.05 as significant.

**Reporting summary.** Further information on research design is available in the Nature Portfolio Reporting Summary linked to this article.

## Data availability
All data supporting the findings of this study are available within the paper and its Supplementary Information. The numerical source data for the graphs can be found in Supplemental Data 1.

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

## Acknowledgements

We thank the Radiation Research Core Laboratory of the Chang Gung Memorial Hospital for the flow cytometry instrument support. The Research Center for Emerging Viral Infections receives support from The Featured Areas Research Center Program within the framework of the Higher Education Sprout Project by the Ministry of Education (MOE), Taiwan and NSTC 111−2634-F-182-001 from the National Science and Technology Council, Taiwan. This work was supported by Projects MOST 111-2314-B-182-029 (A.D.), MOST-105-2321-B-182A-003-MY3 (A.D.), and MOST-106-2314-B-182A-160-MY3 (C.T.H.) from the Ministry of Science & Technology, Taiwan, and by CMRPVVJ0052 (C.Y.H.), CMRPG 3E2081-3, and 3J0721 (C.T.H.) from the Medical Research Project Fund, Chang Gung Memorial Hospital, Taiwan.

## Author contributions

A.D. and C.-T.H. designed the research studies, analyzed the data, and wrote the paper; T.-C.C. performed pathological scoring of lung sections; A.D., C.-Y.H., S.-H.H., C.-S.C., T.-A.C., Y.-C.L. and C.-Y.L. performed the research.

## Competing interests

The authors declare no competing interests.
