## [Peer Review File · Communications Biology]

Reviewers' comments:

Reviewer #1 (Remarks to the Author):

In the manuscript titled "IL-17-EGFR-TRAF4 axis alleviates lung inflammation in severe influenza", the authors use a transgenic HA antigen responsive T-cell receptor mouse model to study the effects of influenza infection in transgenic mice in comparison to WT mice with transferred T-cells from transgenic mice. They have previously shown that the transgenic mice have better infection outcome than WT mice with transferred T-cells. Here they show that control of the infection and the inflammatory response is largely dependent on a robust Th17 response in the transgenic mice, and provide mechanistic details on the regulation of this response. They demonstrate that the Th17 response is mediated by recently emigrated cells from the thymus, and were able to mimic the infusion of thymic cells later in infection by transferring cells at two stages of the infection, which alleviated the severity of disease in WT mice. They showed that aging mice have a progressively weaker Th17 response, which corresponds to progressively more severe disease, and that TGF-beta regulates and is partially required for Th17 response, and that TGF-beta activation is regulated by IL10 and influenza neuraminidase. Finally, they showed that IL-17 mitigation of inflammatory responses is partially dependent on EGFR, likely through binding of IL17 to EGFR and signaling through TRAF4.

This is a good, thorough study. The mechanistic details they provide are considerable, and suggest multiple possible intervention points for mitigating the severe inflammation frequently associated with respiratory diseases.

The fact that IL17 can be both pro- and anti-inflammatory makes these findings particularly interesting, and highlights the complexity involved in the regulation of immunity with infection. It would be helpful for the authors to discuss what is behind the difference in the donor mice vs. WT recipient groups. They mention that they were surprised that by this difference, expecting that the transgenic mice would undergo a more severe response (as would I). Presumably there is some kind of adaptation to having HA-responsive TCRs on all T-cells, natively, but I would like to see some discussion of potential mechanisms. Can we rule out the role of EXTRA cells being introduced in the WT mice, in addition to their native T cells, as playing a role in the increased inflammation of the mice with transferred T-cells? In addition, please comment on how WT mice (without transferred cells) respond, or should be expected to respond to this dose (with regard to inflammation, survival, weight loss, IL17, INFgamma etc.)

It is somewhat difficult to become oriented to the mouse model used in the manuscript. The authors therefore need to clearly differentiate early on the difference between WT mice with transferred transgenic cells and the natively transgenic mice. After clearly explaining the model and the difference between the two groups, they should establish how each will be referred to in the remainder of the manuscript. The information is there, but without familiarity with this model, it is difficult to understand the difference immediately.

Some additional editing of the paper should be undertaken, e.g.

line 65 - 'quells' should be quell

line 69 'well-claimed' ? should be 'well-known'

line 128 'In contrary' should be 'in contrast'

line 325 'derrangement' is not the correct spelling or the correct word, not clear what is meant here.

line 237 First transfer of naïve HA-specific 6.5 CD4+ 237 T cells was. (needs revision)

Reviewer #2 (Remarks to the Author):

The manuscript "IL-17-EGFR-TRAF4 axis alleviates lung inflammation in severe influenza" by Dutta et al. reports an IL-17-EGFR-TRAF4 axis that alleviates lung inflammation in severe influenza in mice. Excessive inflammation may cause severe disease and death in respiratory virus infection. In this manuscript, the authors employed an adoptive transfer model of hemagglutinin (HA)-specific CD4+ T cells from CD4+ TCR-transgenic 6.5 mice into wild type (WT) recipient mice to investigate virus-specific CD4+ T cell responses following influenza viral infection. The authors first demonstrated that such adoptive transfer caused collateral damage and disease aggravation in the recipient mice, although it helped in viral clearance with a predominant IFN- γ -producing Th1

response. However, the donor TCR-tg 6.5 mice did not suffer from severe inflammation or lethality. The authors next found that in TCR-tg 6.5 mice, the initial Th1 response was peaked at day 3 post infection and then waned with time. Interestingly, there was a dominant Th17 response at later stage (days 6 and 9) post infection, which was derived from recent thymic emigrants and served to alleviate inflammation and confer protection against severe influenza in TCR-tg 6.5 mice. The authors verified this finding using several experimental models, including adoptive transfer of IL-17KO HA-specific CD4+ T cells, thymectomy of young TCR-tg 6.5 mice before viral infection, young adult versus aging TCR-tg 6.5 mice, and two sequential transfers of naïve HA-specific CD4+ T cells of TCR-tg 6.5 mice into WT recipient mice at day 0 and day 4 post infection. Mechanistically, the authors revealed that viral neuraminidase-activated TGF- β production in the Th1 cells generated at the initial phase of infection guides the HA-specific Th17 evolution from naïve cells at the later phase of viral infection. Furthermore, the authors provided some evidence suggesting that IL-17 signaling through the non-canonical IL-17 receptor EGFR activated the adaptor protein TRAF4 more than TRAF6 to protect against lung inflammation.

Overall, the studies and findings presented in this manuscript are interesting and significant. However, the evidence supporting the major conclusion on the "IL-17-EGFR-TRAF4 axis" is minimal and not convincing. There are also some concerns regarding the statistical method used in this manuscript and the quality of the flow cytometric data reporting the "second transfer cells" in the experiments of two sequential transfers. In addition, a variety of mismatched color schemes in the figures as well as grammatical errors and typos in the text are identified and need to be corrected.

If the authors would like to keep the major conclusion on the "IL-17-EGFR-TRAF4 axis", the following experiments and data will be critical to improve this manuscript: (1) Does IL-17 induce EGFR phosphorylation in HA-specific CD4+ T cells in vitro upon stimulation and in vivo after viral infection? (2) Does IL-17 up-regulate TRAF4 protein levels, recruit the translocation of TRAF4 to EGFR signaling complex, and activate TRAF4 E3 ligase activity in HA-specific CD4+ T cells in vitro upon stimulation and in vivo after viral infection? (3) How does the EGFR inhibitor Gefitinib affect the IL-17-induced signaling events, including EGFR phosphorylation, TRAF4 protein up-regulation, recruitment of TRAF4 to the EGFR signaling complex, and TRAF4 E3 ligase activity, in HA-specific CD4+ T cells in vitro upon stimulation and in vivo after viral infection? Alternatively, the authors can simply rephrase the title, abstract and text to emphasize other solid findings described in this manuscript and avoid highlighting the "IL-17-EGFR-TRAF4 axis".

Statistics: Student's t-test is the only statistical method used in the entire manuscript. However, this method is neither applicable for the comparison of animal survival curves nor appropriate for the analyses of data with more than 2 groups or cohorts. Statistical methods need to be improved for almost all the figures and supplementary figures.

Revision of figures:

Fig. 1d: There are data of CD8 T cells, but there is no corresponding description in the text and figure legend.

Fig. 4a: There are almost no cells in the "2nd transfer cells" of the two transfers experiment as shown in the bottom right panel of FACS profile at Day 8 post infection, raising the concern that the data of "17.65%" of IL-17 producing cells in the "2nd transfer cells" are not reliable. According to the Methods of this manuscript, 1.5×10^6 TCR tg 6.5 T cells were transferred at the 2nd transfer, while 0.5×10^6 cells were transferred at the 1st transfer. There should be more of the "2nd transfer cells" identified in the FACS profile. If the 1st transfer cells expanded too much and indeed caused a dramatic decrease of the ratio of the 2nd/1st transfer cells at day 8 post infection, the authors need to re-do these experiments with appropriately adjusted cell numbers of the 1st versus 2nd transfer.

Fig. 5a: The IL-17 FACS data are questionable. Although the % of IL-17-producing cells appears to be "24.7%" in the top right panel, the IL-17 staining intensity or production level on the gated IL-17+ cells of the 2nd transfer is much lower than that detected in the 1st transfer cells on a per cell basis. With the striking difference in both the cell number and the production level between the 2nd and 1st transfer cells, it is not convincing that the 2nd transfer cells are the dominant

contributor of IL-17 in the in vivo response to viral infection. Fig. 5c: What do the numbers on the FACS plots mean? This needs to be described in the figure legend. Fig. 5g: Some error bars of the graphs are cut off.

Fig. 7b: Not convincing: although the % of IL-17+ cells appears to be higher (14.3) in the bottom right panel, but the FACS dot profiles raise a concern on the appropriateness of the IL-17 gating line position.

Fig. 8: Neuraminidase is in the title of this figure legend but was not directly examined in this Figure other than the use of HA-inserted Vaccinia for infection. Additional justification?

Fig. 9: Relative dominance of TRAF4 over TRAF6 was only determined at the mRNA level and thus is not convincing. How about the relative dominance of TRAF4 over TRAF6 at protein level and E3 ligase activity level in CD4+ T cells? Fig. 9a: Color schemes are mismatched with the curves. Fig. 9b: Bar graph data of day 3 EGFR in WT mice are not consistent with the representative FACS profile, which shows higher EGFR level on day 3 than in naïve mice.

Fig. 10: These experimental data show the protective effects of IL-17 and EGFR, but not necessarily the EGFR expressed on CD4 T cells. Both EGFR and IL-17RA are also expressed on lung epithelial cells. How could the authors conclude that the effects of the EGFR inhibitor Gefitinib were mediated by CD4+ T cells but not by lung epithelial cells from the in vivo data?

Supplementary Figure 5: What are the red curves? The color schemes are mismatched with the curves.

Minor revisions:

Grammatical errors: for example, Pages 11-12 lines 237-239: "First transfer of naïve HA-specific 6.5 CD4+ T cells was. The cells of both transfers, cells with Vac-HA infection and the cells of second transfer on day 4, were activated into Th1 phenotype, as expected with the absence of active-TGF- β on the cells of first adoptive transfer".

Typos: for example, "inflammed" on page 5 line 97; Fig. 5 legend: "ROR- γ " should be "ROR- γ t"; Supplementary methods: Page 9, labeling of T cells with "5 mM" CFSE, the concentration of CFSE appears to be too high.

Reference citation error on page 15: lines 322-324. Reference citation of the TRAF family is lacking on page 12 lines 254-256.

Ms. Ref. No.: COMMSBIO-22-3077-T

Dear reviewers,

We thank you for reviewing our manuscript. After extensive efforts in these months, we accomplished substantial *in vitro* verification but very minimal with *in vivo* experiments. We stimulated the splenocytes from naïve 6.5 mice, IL-17KO 6.5 mice, or WT mice *in vitro* with 0.03 MOI influenza virus for 48 hours. IL-17 induced EGFR phosphorylation, TRAF4 protein up-regulation, and TRAF4 translocation to the EGFR complex. EGFR inhibitor Gefitinib mitigated these IL-17-induced effects. We are still trying to find a collaborator for the TRAF4 E3 ligase activity study in Taiwan. Due to all the technical difficulties, especially for the availability of mice of desired age and sex and sorting the very limited numbers of cells *in vivo*, we estimated we need much more time for the *in vivo* confirmation.

Mechanistic dissection is very critical here as the first reviewer highlighted that it may offer multiple possible intervention points for mitigating the inflammation associated with severe respiratory illness. We would like to keep a mechanistic title with toned-down wording, as we consider that our *in vitro* verification supports at least partial involvement of the IL-17-EGFR-TRAF4 axis.

We list here our responses to the reviewer's concerns and submit a revised version of the manuscript and figures.

Thanking you.
Best regards,
Ching-Tai Huang.

Comments & Responses

Reviewers' comments:

Reviewer #1 (Remarks to the Author):

In the manuscript titled "IL-17-EGFR-TRAF4 axis alleviates lung inflammation in severe influenza", the authors use a transgenic HA antigen responsive T-cell receptor mouse model to study the effects of influenza infection in transgenic mice in comparison to WT mice with transferred T-cells from transgenic mice. They have previously shown that the transgenic mice have better infection outcome than WT mice with transferred T-cells. Here they show that control of the infection and the inflammatory response is largely dependent on a robust Th17 response in the transgenic mice, and provide mechanistic details on the regulation of this response. They demonstrate that the Th17 response is mediated by recently emigrated cells from the thymus, and were able to mimic the infusion of thymic cells later in infection by transferring cells at two stages of the infection, which alleviated the severity of disease in WT mice. They showed that aging mice have a progressively weaker Th17 response, which corresponds to progressively more severe disease, and that TGF-beta regulates and is partially required for Th17 response, and that TGF-beta activation is regulated by IL10 and influenza neuraminidase. Finally, they showed that IL-17 mitigation of inflammatory responses is partially dependent on EGFR, likely through binding of IL17 to EGFR and signaling through TRAF4. This is a good, thorough study. The mechanistic details they provide are considerable, and suggest multiple possible

intervention points for mitigating the severe inflammation frequently associated with respiratory diseases.

Response: We appreciate the encouraging comments of the reviewer.

The fact that IL17 can be both pro- and anti-inflammatory makes these findings particularly interesting, and highlights the complexity involved in the regulation of immunity with infection. It would be helpful for the authors to discuss what is behind the difference in the donor mice vs. WT recipient groups. They mention that they were surprised that by this difference, expecting that the transgenic mice would undergo a more severe response (as would I). Presumably there is some kind of adaptation to having HA-responsive TCRs on all T-cells, natively, but I would like to see some discussion of potential mechanisms.

Response: Our results showed the recent thymic emigrants of 6.5 transgenic mice or the 2nd batch of naïve cells in WT mice with two adoptive transfers were activated and evolved into regulatory T17 cells, in the presence of Th1 effector cells. In WT type with only one adoptive transfer, there were no new naïve T cells which may evolve into regulatory phenotype.

Can we rule out the role of EXTRA cells being introduced in the WT mice, in addition to their native T cells, as playing a role in the increased inflammation of the mice with transferred T-cells? In addition, please comment on how WT mice (without transferred cells) respond, or should be expected to respond to this dose (with regard to inflammation, survival, weight loss, IL17, INFgamma etc.)

Response: WT mice without extra cells also suffer from infection with BW loss and mortality. Extra cells of adoptive transfer enhance the BW loss and increase the mortality with increased IFN- γ and IL-17 production and inflammation. We have shown this in our previous publications (Dutta A, et al., *Journal of Immunology*, 2013, 190:4205-4214; Dutta A, et al., *Nature Communications*, 2015, 6:6374).

It is somewhat difficult to become oriented to the mouse model used in the manuscript. The authors therefore need to clearly differentiate early on the difference between WT mice with transferred transgenic cells and the natively transgenic mice. After clearly explaining the model and the difference between the two groups, they should establish how each will be referred to in the remainder of the manuscript. The information is there, but without familiarity with this model, it is difficult to understand the difference immediately.

Response: We specified wild-type mice with transferred transgenic cells as “WT mice + 6.5 cells” and natively transgenic mice as “6.5 mice”. Accordingly, we specified IL-17KO mice with transferred IL-17KO transgenic cells as “IL-17KO mice + IL-17KO 6.5 cells” and natively IL-17KO transgenic mice as “IL-17KO 6.5 mice”. We made this nomenclature all the same throughout the revised manuscript.

Some additional editing of the paper should be undertaken, e.g. line 65 - 'quells' should be quell

Response: We apologize for the typo, and amended the text (line 59 of revised manuscript), as suggested.

line 69 'well-claimed' ? should be 'well-known'

Response: We amended the text (lines 63-64 of revised manuscript), as suggested.

line 128 'In contrary' should be 'in contrast'

Response: We amended the text (line 124 of revised manuscript), as suggested.

line 325 'derrangement' is not the correct spelling or the correct word, not clear what is meant here.

Response: We corrected the typo and amended the text as “derangement”, line 359 of revised manuscript.

line 237 First transfer of naïve HA-specific 6.5 CD4+ 237 T cells was. (needs revision)

Response: We deleted this part, line 237 of revised manuscript.

Reviewer #2 (Remarks to the Author):

The manuscript “IL-17-EGFR-TRAF4 axis alleviates lung inflammation in severe influenza” by Dutta et al. reports an IL-17-EGFR-TRAF4 axis that alleviates lung inflammation in severe influenza in mice. Excessive inflammation may cause severe disease and death in respiratory virus infection. In this manuscript, the authors employed an adoptive transfer model of hemagglutinin (HA)-specific CD4+ T cells from CD4+ TCR-transgenic 6.5 mice into wild type (WT) recipient mice to investigate virus-specific CD4+ T cell responses following influenza viral infection. The authors first demonstrated that such adoptive transfer caused collateral damage and disease aggravation in the recipient mice, although it helped in viral clearance with a predominant IFN- γ -producing Th1 response. However, the donor TCR-tg 6.5 mice did not suffer from severe inflammation or lethality. The authors next found that in TCR-tg 6.5 mice, the initial Th1 response was peaked at day 3 post infection and then waned with time. Interestingly, there was a dominant Th17 response at later stage (days 6 and 9) post infection, which was derived from recent thymic emigrants and served to alleviate inflammation and confer protection against severe influenza in TCR-tg 6.5 mice. The authors verified this finding using several experimental models, including adoptive transfer of IL-17KO HA-specific CD4+ T cells, thymectomy of young TCR-tg 6.5 mice before viral infection, young adult versus aging TCR-tg 6.5 mice, and two sequential transfers of naïve HA-specific CD4+ T cells of TCR-tg 6.5 mice into WT recipient mice at day 0 and day 4 post infection. Mechanistically, the authors revealed that viral neuraminidase-activated TGF- β production in the Th1 cells generated at the initial phase of infection guides the HA-specific Th17 evolution from naïve cells at the later phase of viral infection. Furthermore, the authors provided some evidence suggesting that IL-17 signaling through the non-canonical IL-17 receptor EGFR activated the adaptor protein TRAF4 more than TRAF6 to protect against lung inflammation.

Overall, the studies and findings presented in this manuscript are interesting

and significant. However, the evidence supporting the major conclusion on the “IL-17-EGFR-TRAF4 axis” is minimal and not convincing. There are also some concerns regarding the statistical method used in this manuscript and the quality of the flow cytometric data reporting the “second transfer cells” in the experiments of two sequential transfers. In addition, a variety of mismatched color schemes in the figures as well as grammatical errors and typos in the text are identified and need to be corrected.

If the authors would like to keep the major conclusion on the “IL-17-EGFR-TRAF4 axis”, the following experiments and data will be critical to improve this manuscript: (1) Does IL-17 induce EGFR phosphorylation in HA-specific CD4⁺ T cells *in vitro* upon stimulation and *in vivo* after viral infection?

Response:

(A) *In vitro* verification: We stimulated splenocytes from naïve 6.5 mice, IL-17KO 6.5 mice, or WT mice with 0.03 MOI influenza virus for 48 hours. Cells were re-stimulated with cognate HA peptide in the presence of brefeldin-A for last 3 hours. We purchased FITC-tagged Phospho-EGFR (Tyr1068) Recombinant Rabbit Monoclonal Antibody (EGFRY1068-E5; Cat# MA5-27995) from Invitrogen™. Flow cytometry revealed increased EGFR phosphorylation in the HA-specific CD4⁺ T cells from IL-17-competent 6.5 mice than the IL-17KO 6.5 mice. IL-17 supplement augmented EGFR phosphorylation of the HA-specific CD4⁺ T cells from IL-17KO mice in a dose-dependent manner, more with 1.0 than 0.1 µg IL-17.

(B) *In vivo* verification: Single cell suspensions from lungs of infected 6.5 transgenic mice were re-stimulated *in vitro* with cognate HA peptide. Brefeldin-A was added for last 3 hours of re-stimulation. Flow cytometry revealed more EGFR phosphorylation on day 9 than day 6 or day 3. In contrary, there was not such increase in the absence of IL-17, as in the adoptively transferred IL-17KO HA-specific 6.5 CD4⁺ T cells in the lungs of infected IL-17KO mice.

(2) Does IL-17 up-regulate TRAF4 protein levels, recruit the translocation of TRAF4 to EGFR signaling complex, and activate TRAF4 E3 ligase activity in HA-specific CD4⁺ T cells in vitro upon stimulation and in vivo after viral infection?

Response:

(A) *In vitro* verification:

1. TRAF4 protein levels: We stimulated splenocytes from naïve 6.5 mice, IL-17KO 6.5 mice, or WT mice with 0.03 MOI influenza virus for 48 hours. Cells were re-stimulated with cognate HA peptide in the presence of brefeldin-A for last 3 hours. We purchased PE-tagged TRAF4 Antibody (B-9, mouse monoclonal IgG₁ κ; Cat# Sc-390232) from Santa Cruz Biotechnology. Flow cytometry revealed more TRAF-4 in the HA-specific CD4⁺ T cells from 6.5 mice than the IL-17KO 6.5 mice. IL-17 supplement augmented intracellular TRAF-4 in cells from IL-17KO mice in a dose-dependent manner, more with 1.0 than 0.1 μg IL-17.

2. Translocation of TRAF4 to EGFR signaling complex: We stimulated splenocytes from naïve 6.5 mice with 0.03 MOI influenza virus for 48 hours, and re-stimulated the cells with cognate HA peptide in the presence of brefeldin-A for last 3 hours. Confocal microscopy revealed TRAF4 translocation to EGFR signaling complex. In contrary, there was less translocation in the absence of IL-17, as in the cells from IL-17KO 6.5 mice.

3. TRAF4 E3 ligase activity: We do not have expertise of TRAF4 E3 ligase activity study. We are still trying to find a collaborator to do this experiment.

(B) *In vivo* verification: We are preparing the experiments and do not have substantial results yet due to technical difficulties, especially for the availabilities of the mice needed for multiple adoptive transfer and extensive sorting. Shortage is always a problem as we breed these mice by our own in our animal facility.

(3) How does the EGFR inhibitor Gefitinib affect the IL-17-induced signaling events, including EGFR phosphorylation, TRAF4 protein up-regulation, recruitment of TRAF4 to the EGFR signaling complex, and TRAF4 E3 ligase activity, in HA-specific CD4 +T cells *in vitro* upon stimulation and *in vivo* after viral infection?

Response: In the experiments mentioned above, Gefitinib suppressed the IL-17-induced signaling events, including EGFR phosphorylation, TRAF4 protein up-regulation, and recruitment of TRAF4 to the EGFR signaling complex. We have not checked TRAF4 E3 ligase activity yet. Gefitinib added in the cultures was 1.0 μg as reported in literature (doi:10.1186/1471-2407-4-83; doi.org/10.1158/1535-7163.MCT-08-1219). We are preparing the *in vivo* experiments and do not have substantial results yet due to technical difficulties we mentioned above.

Alternatively, the authors can simply rephrase the title, abstract and text to emphasize other solid findings described in this manuscript and avoid highlighting the “IL-17-EGFR-TRAF4 axis”.

Response: Mechanistic dissection is the critical part of our research as it may provide multiple possible intervention points for mitigating the inflammation associated with severe respiratory illness. We would like to keep a mechanistic title with toned-down wording as our *in vitro* verification supports at least partial involvement of the IL-17-EGFR-TRAF4 axis. We amended to tone-down:

- (1) **Title:** IL-17-EGFR-TRAF4 axis contributes to the alleviation of lung inflammation in severe influenza.
- (2) **Abstract:** Added phrase: “Our results suggest”.

(3) Result section titles:

- a. From “IL-17–EGFR-TRAF4 axis and the alleviation of lung inflammation in severe influenza” to “IL-17 signaling through the non-canonical IL-17 receptor EGFR activates the scaffold protein TRAF4 more than TRAF6 in the lung-infiltrating HA-specific CD4+ T cells”
- b. From “Disrupted IL-17–EGFR-TRAF4 axis with Gefitinib treatment aggravates the disease of severe influenza” to “Treatment with EGFR inhibitor, Gefitinib, aggravates the disease of severe influenza”.

Statistics: Student’s t-test is the only statistical method used in the entire manuscript. However, this method is neither applicable for the comparison of animal survival curves nor appropriate for the analyses of data with more than 2 groups or cohorts. Statistical methods need to be improved for almost all the figures and supplementary figures.

Response: We have comparison of 2 groups only. We added Two-way ANOVA for body weight curves and Gehan-Breslow-Wilcoxon test for survival curves in addition to the previously done Student’s t-test for comparison between 2 groups on a specific day in the revised manuscript.

Revision of figures:

Fig. 1d: There are data of CD8 T cells, but there is no corresponding description in the text and figure legend.

Response: We added the corresponding description in the text and figure legend, as suggested (lines 117 & 535 of revised manuscript).

Fig. 4a: There are almost no cells in the “2nd transfer cells” of the two transfers experiment as shown in the bottom right panel of FACS profile at Day 8 post infection, raising the concern that the data of “17.65%” of IL-17 producing cells in the “2nd transfer cells” are not reliable. According to the Methods of this manuscript, 1.5×10^6 TCR tg 6.5 T cells were transferred at the 2nd transfer, while 0.5×10^6 cells were transferred at the 1st transfer. There should be more of the “2nd transfer cells” identified in the FACS profile. If the 1st transfer cells expanded too much and indeed caused a dramatic decrease of the ratio of the 2nd/1st transfer cells at day 8 post infection, the authors need to re-do these experiments with appropriately adjusted cell numbers of the 1st versus 2nd transfer.

Response: We repeat the experiment and take dot plots with substantially more events. The 2nd batch cells expanded under the influence of the 1st batch cells. Expansion of the 1st batch cells may have caused space problem and limit the expansion of the 2nd batch cells. We keep using the cell number set in this defined experiment, not to add in one more confounding factors for consistency and reproducibility.

Fig. 5a: The IL-17 FACS data are questionable. Although the % of IL-17-producing cells appears to be "24.7%" in the top right panel, the IL-17 staining intensity or production level on the gated IL-17+ cells of the 2nd transfer is much lower than that detected in the 1st transfer cells on a per cell basis. With the striking difference in both the cell number and the production level between the 2nd and 1st transfer cells, it is not convincing that the 2nd transfer cells are the dominant contributor of IL-17 in the *in vivo* response to viral infection.

Response: We could not exclude the possibility that certain amount of IL-17 was contributed by the 1st batch cells in this context. Even the IL-17 per cell was lower for the 2nd than the 1st batch, experiments with adoptive transfer of IL-17KO cells implied that the IL-17 from the 2nd batch cells is crucial for the relief of inflammation.

Fig. 5c: What do the numbers on the FACS plots mean? This needs to be described in the figure legend. Fig. 5g: Some error bars of the graphs are cut off.

Response: The numbers are MFI (Mean Fluorescence intensity), and the information is now added in the figure legend. We also corrected the cut off error bars of Fig. 5g in the revised manuscript, as shown below:

Fig. 7b: Not convincing: although the % of IL-17+ cells appears to be higher (14.3) in the bottom right panel, but the FACS dot profiles raise a concern on the appropriateness of the IL-17 gating line position.

Response: We decided the position of the line according to FMO (Fluorescence Minus One) control. Even if the gating line was moved further to the right, the difference was still there. We saw a bigger difference with repeated experiment for a longer stimulation *in vitro* (48 hours instead of overnight).

Fig. 8: Neuraminidase is in the title of this figure legend but was not directly examined in this Figure other than the use of HA-inserted Vaccinia for infection. Additional justification?

Response: We demonstrated viral neuraminidase-mediated activation of TGF- β in the lung-infiltrating HA-specific CD4⁺ T cells in our previous research article (Dutta A, et al., *Nature Communications*, 2015, 6:6374) but not here. We changed the title of the figure legend as “Influenza virus-activated TGF- β of the Th1 cells guides Th17 response of naïve cognate antigen-specific CD4⁺ T cells” (lines 619-620 of revised manuscript).

Fig. 9: Relative dominance of TRAF4 over TRAF6 was only determined at the mRNA level and thus is not convincing. How about the relative dominance of TRAF4 over TRAF6 at protein level and E3 ligase activity level in CD4⁺ T cells?

Response: We could not get TRAF6 antibody tagged with a colour different from that of anti-TRAF4 to fit into our multicolor staining panel. We cannot exercise the western-blot analyses which required a huge number of cells very difficult to get with our in vivo adoptive transfer system.

Fig. 9a: Color schemes are mismatched with the curves. Fig. 9b: Bar graph data of day 3 EGFR in WT mice are not consistent with the representative FACS profile, which shows higher EGFR level on day 3 than in naïve mice.

Response: We corrected the Color schemes and bar graph data as shown below:

Fig. 10: These experimental data show the protective effects of IL-17 and EGFR, but not necessarily the EGFR expressed on CD4 T cells. Both EGFR and IL-17RA are also expressed on lung epithelial cells. How could the authors conclude that the effects of the EGFR inhibitor Gefitinib were mediated by CD4⁺ T cells but not by lung epithelial cells from the in vivo data?

Response: The CD4⁺ cells are the target of our antigen specific investigation. We did not exclude the possible contribution of other cells, such the epithelial cells in the lung.

Supplementary Figure 5: What are the red curves? The color schemes are mismatched with the curves.

Response: The red curves are the data from 6.5 mice. We corrected and matched the color schemes in the revised manuscript, as shown below:

Minor revisions:

Grammatical errors: for example, Pages 11-12 lines 237-239: “First transfer of naïve HA-specific 6.5 CD4⁺ T cells was. The cells of both transfers, cells with Vac-HA

infection and the cells of second transfer on day 4, were activated into Th1 phenotype, as expected with the absence of active-TGF- β on the cells of first adoptive transfer”.

Response: We corrected this part as: “The donor 6.5 cells of both transfers, transferred with infection and transferred on day 4, were activated into Th1 phenotype by the Vac-HA infection, as expected with the absence of active-TGF- β on the first transfer donor 6.5 cells.” (Lines 237-239 of the revised manuscript).

Typos: for example, “inflammed” on page 5 line 97; Fig. 5 legend: “ROR- γ ” should be “ROR- γ t”; Supplementary methods: Page 9, labeling of T cells with “5 mM” CFSE, the concentration of CFSE appears to be too high.

Response: We apologize for the typo. We corrected them as “inflamed” (Line 93 of the revised manuscript), “ROR- γ t” (Lines 578, 580, & 585 of the revised manuscript) and “5 μ M” (Page 9, Supplementary methods) in the revised manuscript.

Reference citation error on page 15: lines 322-324. Reference citation of the TRAF family is lacking on page 12 lines 254-256.

Response: We corrected the citation error, as: “pandemics^{44, 45}, <https://www.cdc.gov/coronavirus/2019-ncov/need-extra-precautions/index.html>; accessed August 23, 2022.” We also added a reference citation of the TRAF family (*Ref: 28*), as: “Bradley, J. R., Pober, J. S. Tumor necrosis factor receptor-associated factors (TRAFs). *Oncogene* **20**, 6482-91 (2001).”

Sincerely,

Ching-Tai Huang, M.D., Ph. D.
Division of Infectious Diseases, Department of Medicine,
Chang Gung Memorial Hospital,
Taoyuan, Taiwan.
Phone: +886-3-3281200 Ext: 8740; Fax: +886-3-3289410;
Email: chingtaihuang@gmail.com

REVIEWERS' COMMENTS:

Reviewer #1 (Remarks to the Author):

I am satisfied with the authors' responses to my comments.

Reviewer #2 (Remarks to the Author):

The authors have addressed my previous concerns.

Ms. Ref. No.: COMMSBIO-22-3077B

Dear Dr. Christina Karlsson Rosenthal,
Chief Editor, Communications Biology.

We thank you and both the reviewers for considering our manuscript for publication in your journal.

Both the reviewers are satisfied with our responses to their previous concerns, and no further concern remains for this manuscript.

Thanking you.
Best regards,
Ching-Tai Huang.

Comments & Responses

Reviewers' comments:

Reviewer #1 (Remarks to the Author):

I am satisfied with the authors' responses to my comments.

Response: We appreciate the encouraging comments of the reviewer.

Reviewer #2 (Remarks to the Author):

The authors have addressed my previous concerns.

Response: We appreciate the encouraging comments of the reviewer.

Sincerely,

Ching-Tai Huang, M.D., Ph. D.
Division of Infectious Diseases, Department of Medicine,
Chang Gung Memorial Hospital,
Taoyuan, Taiwan.
Phone: +886-3-3281200 Ext: 8740; Fax: +886-3-3289410;
Email: chingtaihuang@gmail.com